# STK-12 acts as a transcriptional brake to control the expression of cellulase-encoding genes in *Neurospora crassa*

Liangcai Lin[1], Shanshan Wang[1], Xiaolin Li[1,2], Qun He[2], J. Philipp Benz[3,4], Chaoguang Tian[1]*

**1** Key Laboratory of Systems Microbial Biotechnology, Tianjin Institute of Industrial Biotechnology, Chinese Academy of Sciences, Tianjin, China, **2** State Key Laboratory of Agrobiotechnology and MOA Key Laboratory of Soil Microbiology, College of Biological Sciences, China Agricultural University, Beijing, China, **3** Technical University of Munich, TUM School of Life Sciences Weihenstephan, Hans-Carl-von-Carlowitz-Platz, Freising, Germany, **4** Technical University of Munich, Institute for Advanced Study, Lichtenbergstr, Garching, Germany

\* tian_cg@tib.cas.cn

**Data Availability Statement:** All relevant data are within the manuscript and its Supporting Information files.

## Abstract

Cellulolytic fungi have evolved a complex regulatory network to maintain the precise balance of nutrients required for growth and hydrolytic enzyme production. When fungi are exposed to cellulose, the transcript levels of cellulase genes rapidly increase and then decline. However, the mechanisms underlying this bell-shaped expression pattern are unclear. We systematically screened a protein kinase deletion set in the filamentous fungus *Neurospora crassa* to search for mutants exhibiting aberrant expression patterns of cellulase genes. We observed that the loss of *stk-12* (NCU07378) caused a dramatic increase in cellulase production and an extended period of high transcript abundance of major cellulase genes. These results suggested that *stk-12* plays a critical role as a brake to turn down the transcription of cellulase genes to repress the overexpression of hydrolytic enzymes and prevent energy wastage. Transcriptional profiling analyses revealed that cellulase gene expression levels were maintained at high levels for 56 h in the Δ*stk-12* mutant, compared to only 8 h in the wild-type (WT) strain. After growth on cellulose for 3 days, the transcript levels of cellulase genes in the Δ*stk-12* mutant were 3.3-fold over WT, and *clr-2* (encoding a transcriptional activator) was up-regulated in Δ*stk-12* while *res-1* and *rca-1* (encoding two cellulase repressors) were down-regulated. Consequently, total cellulase production in the Δ*stk-12* mutant was 7-fold higher than in the WT. These results strongly suggest that *stk-12* deletion results in dysregulation of the cellulase expression machinery. Further analyses showed that STK-12 directly targets IGO-1 to regulate cellulase production. The TORC1 pathway promoted cellulase production, at least partly, by inhibiting STK-12 function, and STK-12 and CRE-1 functioned in parallel pathways to repress cellulase gene expression. Our results clarify how cellulase genes are repressed at the transcriptional level during cellulose induction, and highlight a new strategy to improve industrial fungal strains.

**Funding:** The work was supported by the National Natural Science Foundation of China(NSFC): 31761133018(CT) and 31501007(LL), The National Key Program of Research and Development (2018YFA0900500, CT), Chinese Academy of Sciences (XDA21060900, CT), and the Deutsche Forschungsgemeinschaft (DFG, German Research Foundation): grant BE 6069/3-1 (JPB). The funders had no role in study design, data collection and analysis, decision to publish, or preparation of the manuscript.

**Competing interests:** The authors have declared that no competing interests exist.

## Author summary

Microorganisms can sense and respond to nutrient availability in the external environment, and turn on/off cellular signaling pathways to control gene expression in a timely manner. In filamentous fungi, the expression of hydrolytic enzymes is tightly controlled at the transcriptional level. Within fungal cells, signals from induction and repression pathways are integrated, resulting in optimal hydrolase gene expression. However, the detailed molecular mechanism of transcriptional down-regulation of hydrolytic enzyme genes remains poorly understood. The filamentous fungus *Neurospora crassa*, a native degrader of lignocellulosic biomass, was developed as a model to unravel mechanisms of lignocellulolytic gene regulation. Using this system, we systematically screened *N. crassa* serine-threonine protein kinase mutants by determining their cellulase production capacity and identified STK-12 to be a crucial factor for cellulase gene downregulation. Since STK-12 is conserved across species, our data potentially provide insights into the repression mechanism that controls hydrolase gene expression also in other filamentous fungi, and will be useful in the rational engineering of fungal strains to improve industrial enzyme production.

## Introduction

Fine-tuning of gene expression is essential for all organisms. Complex regulatory networks ensure that the magnitude and duration of gene expression are accurately controlled, and that gene expression is turned off in a timely manner. For instance, in humans, the expression levels of immune response genes rapidly increase upon immune challenge [1]. However, sustained expression of these genes can result in tissue damage and immune pathologies [2]. Also in lower eukaryotes, defects in the regulation of gene expression are disadvantageous for cell survival [3].

Cellulolytic fungi, such as *Trichoderma reesei* and *Neurospora crassa*, play important roles in ecosystems, and are responsible for a major part of plant biomass degradation [4,5]. To survive under lignocellulolytic conditions, these fungi have evolved a remarkable capability to secret cellulolytic enzymes that convert insoluble polysaccharides into fermentable sugars. However, the production of excessive amounts of hydrolytic enzymes is detrimental for cell survival because too much cellular resources are directed towards protein synthesis. Previous transcriptome profiling studies have demonstrated that genes encoding major cellulases exhibit a typical bell-shaped expression pattern, implying that fungi produce cellulolytic enzymes in a strictly controlled manner [4]. The cellulase gene expression pattern can be divided into two stages: Initial sensing and induction (Stage I) followed by repression (Stage II). Many studies have focused on Stage I so far. For example, it has been demonstrated that the cellulose response transporter CRT1 is required for the transcriptional response to sophorose, a principal inducer of cellulase production in *T. reesei* [6]. Similarly, two *N. crassa* cellodextrin transporters, CDT-1 and CDT-2, play essential roles in the sensing and uptake of cellulase inducers [7,8]. Other studies have identified and characterized multiple critical transcription factors involved in the regulation of cellulase genes, such as the activators XYR1 [9], CLR-1/2 [10], CLR-4 [11], and ACE3 [12]. Whereas many studies have focused on Stage I, fewer have focused on Stage II, *i.e.* how cellulase gene expression is repressed. Previous studies have shown that, when *N. crassa* is cultured in an inducing medium, the expression levels of cellulase genes dramatically increase at early time points and then rapidly decline [4,13]. Similarly, the transcriptional down-regulation of genes encoding secreted enzymes has also been

observed in *T. reesei* [14] and *Aspergillus niger* [15], where it was designated as repression under secretion stress (RESS). Although this transcriptional regulation has been proven to be mediated by cellulase gene promoters, the detailed molecular mechanism of RESS is still largely unknown. Deciphering the mechanisms underlying this bell-shaped gene expression pattern and identification of novel signaling pathways involved in transcriptional down-regulation of cellulase genes will increase our understanding of how cellulase genes are regulated, and provide promising avenues for enhancing lignocellulase production in commercial fungal strains.

For cellulolytic fungi, insoluble polysaccharides serve as non-preferred carbon sources. Thus, besides the cellulase induction pathway, nutrient signaling pathways, such as the carbon catabolite repression (CCR) pathway [16], are likely to be involved in the regulation of cellulase gene expression. Fungal cells sense and respond to nutrients through different signaling pathways [17]. Protein kinases, which are integral components of these signaling pathways, transmit signals to their downstream targets and play crucial roles in the regulation of virtually all cellular processes, including metabolism, morphogenesis, and autophagy [18,19]. Several kinase cascades involved in regulating cellulolytic enzymes have been reported in the last decade. For instance, three *T. reesei* mitogen-activated protein (MAP) kinases, Tmk1, Tmk2, and Tmk3, are involved in cellulase formation by different mechanisms. The deletion of *tmk3* in *T. reesei* led to severe defects in protein production and cellulolytic enzyme activities due to down-regulation of cellulase genes [20]. In contrast, the deletion of *tmk2* in *T. reesei* caused increased protein secretion, probably due to a defect in cell wall integrity [21]. Tmk1 was shown to decrease cellulase production by repressing cellular growth in the presence of wheat bran and Avicel [22]. Recent work demonstrated that the MAPKK kinase Ste20, the upstream component of the Tmk3 cascade, is necessary for cellulase gene transcription [23]. In addition, it has been reported that the Ime2-like MAPK negatively modulates the transcript levels of cellulase genes and is required for the expression of XYR1 and CRE1 in *T. reesei* [24].

The cyclic AMP (cAMP)-dependent protein kinase A (PKA), a crucial component of the cAMP pathway, also affects cellulase gene expression through different signal transduction pathways. Schuster *et al.* revealed that PKA is involved in modulating cellulase gene transcription in response to light and functions as a key element in the integrated light and nutrient signaling cascades [25]. Deletion of *pkaA* in *Aspergillus nidulans* resulted in carbon catabolite repression (CCR) mis-function, and caused increased cellulase production [26]. Other protein kinases, such as Sch9/SchA, Yak1 and SnfA, have also been proven to affect cellulase formation [27,28]. Together, these results strongly suggest that protein kinases play critical roles in regulating cellulase gene expression. However, despite these advances, the majority of components of intercellular signaling pathways involved in the regulation of cellulase gene expression still await characterization.

The model filamentous fungus *N. crassa* has been developed as an efficient system for dissecting the mechanisms of cellulase gene regulation because of its natural capacity to secrete enzymes involved in lignocellulose utilization. Previous studies on *N. crassa* have yielded a range of genetic techniques and tools [29,30], including a nearly complete single-gene deletion strain collection (http://www.fgsc.net/ncrassa.html). Thus, large-scale screening of kinase mutants in *N. crassa* is easy to achieve [31,32]. Exploiting these advantages, we screened a library of 64 serine/threonine protein kinase deletion mutants to further unravel the molecular mechanisms and pathways governing cellulase expression in this filamentous fungus. Several candidate genes affected the cellulase titer when disrupted in *N. crassa*. Intriguingly, the disruption of *stk-12* (NCU07378), the homolog of *Saccharomyces cerevisiae rim15*, led to a dramatic change in cellulase production. In *S. cerevisiae*, RIM15 plays critical roles in the response to nutrient status, and integrates nutrient signals coming from different nutrient-sensing

pathways, such as the PKA and Sch9 pathways [33]. Previous studies have demonstrated that RIM15 is negatively regulated by PKA through phosphorylation at five conserved sites (RRXS) [34] and Sch9 is likely to regulate RIM15 function via controlling its subcellular localization [35]. Loss of RIM15 was shown to severely reduce the viability of cells under nutrient-limited conditions [33]. However, reduced viability due to deletion of *rim15* may be desirable in some cases. Recent comparative genomic investigations revealed that RIM15 plays a role in controlling alcoholic fermentation [36]. Deletion of *rim15* significantly increased the fermentation rate and shortened the fermentation period [37]. Recent studies have revealed that Rim15 homologs are important for pathogenicity in *Magnaporthe oryzae* and *Fusarium graminearum* [38,39]. The loss of Rim15 homologs in *Aspergillus spp.* was shown to reduce stress tolerance [40,41]. Furthermore, recent studies revealed that Igo-1/2 and PP2A are critical components of the Rim15 pathway in *S. cerevisiae* [37]. Igo-1 and Igo-2, which are critical for initiation of the $G_0$ program, are directly activated by Rim15 through phosphorylation [42]. The activated IGOs can transmit signals to downstream effectors by inhibition PP2A [43]. The orthologues pathway has also been found in *Schizosaccharomyces pombe*, and this pathway plays a pivotal role in carbohydrate metabolism [36]. Despite the advances made in those studies, the mechanism of how the STK-12 pathway affects cellulase production has not been described.

In this work, we report that STK-12 functions as a novel transcriptional brake that curbs cellulase gene expression in *N. crassa*. The *stk-12* disrupted strain not only showed enhanced cellulase production but also retained high expression levels of cellulase genes for longer than the wild-type (WT) strain. We demonstrate that IGO-1 (NCU03860), which functions as a critical component of the STK-12 pathway, directly interacts with, and is activated by, STK-12 through phosphorylation on Ser-47. We also find PP2A in *N. crassa* (NCU06563) to be a downstream effector that plays a critical role in the STK-12-mediated signaling pathway. Furthermore, our data indicate that STK-10, an important element of the TORC1 signaling pathway, affects cellulase expression at least in part via the STK-12 pathway. Finally, we also demonstrate that CRE-1 and the STK-12 pathway function in parallel pathways to regulate cellulase gene expression. These findings provide new insights into how filamentous fungi regulate cellulase expression and offer novel strategies for engineering of industrial strains.

## Results

### Screening of mutants

To identify critical factors involved in turning down the expression of cellulase genes, the deletion mutants of the 64 genes in the *N. crassa* genome that encode serine-threonine protein kinases were screened through batch culturing with crystalline cellulose (Avicel PH-101) as the sole carbon source. Their cellulase production capacities are shown in Fig 1A and S1 Table. Compared with the WT strain, 15 mutants showed significant changes in secreted protein levels, and seven of these mutants had markedly increased secretion of cellulases. These mutants included *prk-10* (NCU04566), whose deletion increased cellulase production by approximately 34%, *prk-2* (NCU07872), whose deletion enhanced protein secretion by more than 35%, and *stk-12* (NCU07378), whose deletion increased lignocellulase production by 138%. It should be noted that, in *S. cerevisiae*, homologs of these genes are involved in nutrient signal transduction, implying a close link between the cellulase regulatory network and nutrient response pathways. Strains carrying a deletion of Δ*stk-36* (NCU05658) or Δ*stk-46* (NCU06638) also showed markedly increased levels of secreted protein (>70%).

To further explore the roles of these kinase genes in cellulase gene transcription, the kinetics of cellulase gene expression in these hyper-production mutants were determined by quantitative reverse-transcriptase polymerase chain reaction (RT-qPCR) analyses. In the WT strain

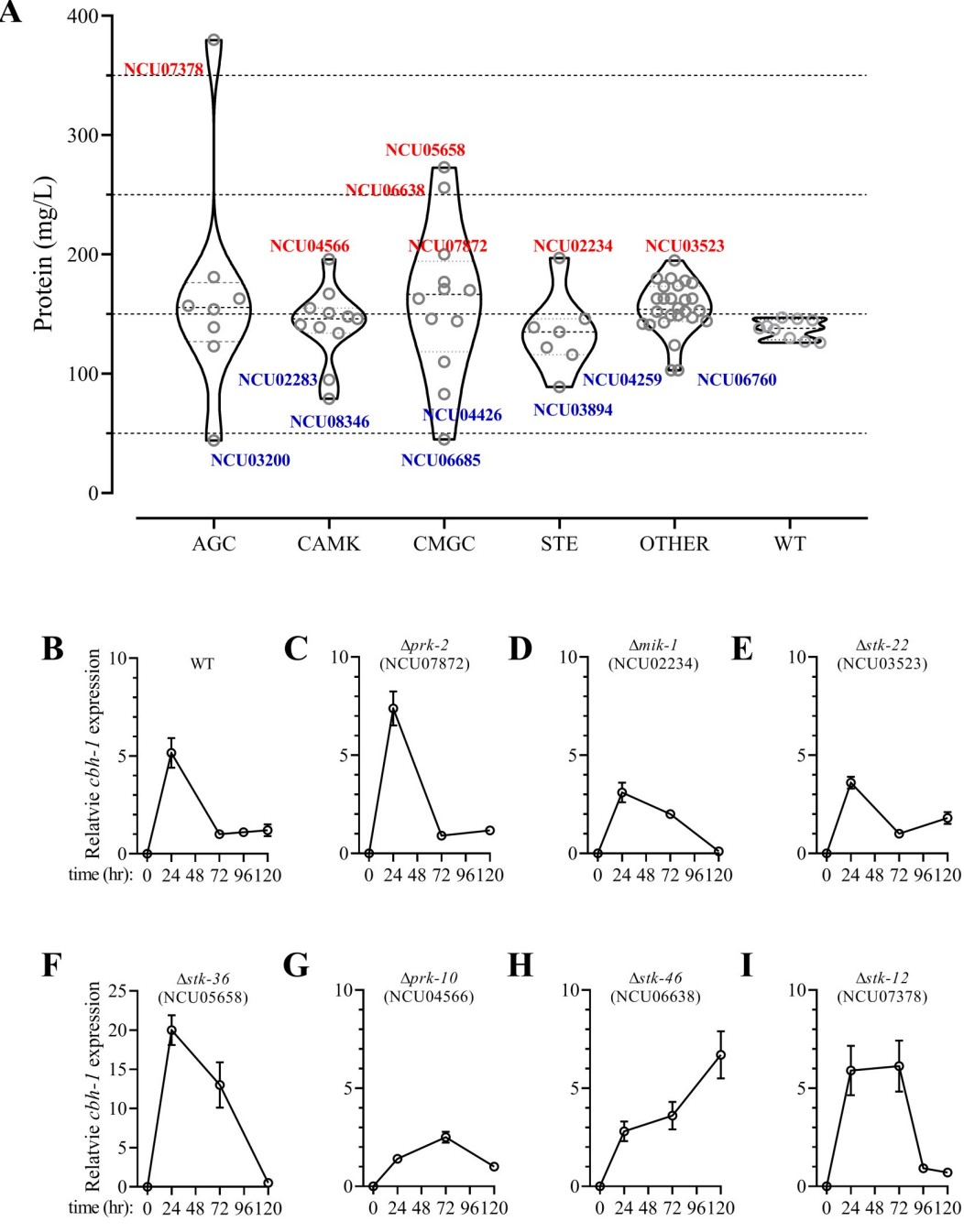

**Fig 1. Screening of serine/threonine protein kinase mutants of *Neurospora crassa*.** (A) Total secreted protein assay was performed on 64 deletion strains as compared to wild-type (WT) strain. Conidia from WT and kinase mutants were separately inoculated into Avicel medium and batch cultured for 5 days. Strains with altered cellulase production (>25%) are indicated as follows: Red, hyper-producing strains; Blue, hypo-producing strains. Serine/threonine protein kinases can be grouped into different families based on catalytic domains: AGC, CAMK, STE, CMGC, and OTHER. (B)-(I) Relative transcript levels of major cellulase gene (*cbh-1*) in hyper-producing mutants versus WT after 1–5 days growth on Avicel. After growth of conidia in Avicel for 24, 72, or 120 h, transcript abundance of *cbh-1* was evaluated by quantitative real-time PCR. Data are normalized against tested gene's transcript level in WT strain at 72 h. *Actin* (NCU04173) transcript levels were used as endogenous control in all samples. Values represent means of at least three biological replicates; error bars show standard deviation.

and the majority of hyper-production mutants, the transcription of the cellulase gene *cbh-1* was significantly induced at early time points (0–24 h) but rapidly declined thereafter (Fig 1B– 1F). This phenomenon was consistent with previous observations [4]. The transcript level of *cbh-1* gradually increased in the Δ*prk-10* mutant and peaked at 72 h (Fig 1G), suggesting that the loss of *prk-10* blunted the response to cellulose. Surprisingly, *cbh-1* showed a completely different expression profile in the Δ*stk-46* mutant. The transcript abundance of *cbh-1* continued to increase during incubation (Fig 1H), which may explain the hyper-production phenotype of this mutant.

Interestingly, the deletion of *stk-12* delayed the repression of *cbh-1* transcription. Although *cbh-1* showed a typical bell-shaped expression pattern in the Δ*stk-12* mutant, the transcript levels were maintained at the peak level until 72 h (Fig 1I). Furthermore, the transcript level of *stk-12* was significantly up-regulated 3.6-fold under lignocellulolytic conditions (S1 Fig), suggesting that *stk-12* transcription was stimulated under nutrient-limited conditions. On the basis of these findings, Δ*stk-12* was chosen for further investigation. This mutation resulted in clear phenotypes in our screening analyses, indicating that its non-mutated gene probably plays an important role as a brake to turn down the transcription of cellulase genes.

## STK-12 is a novel negative regulator involved in cellulase biosynthesis

We conducted a detailed analysis of the effects of *stk-12* disruption on cellulase production. Conidia of WT and Δ*stk-12* were inoculated into 2% Avicel Vogel's minimal medium (VMM) and batch cultured for 5 days. After 3 days of culture, Δ*stk-12* had approximately 7-fold higher lignocellulolytic enzyme production than the WT strain (Fig 2A). Consistent with the increased protein level, the activities of endoglucanase, exoglucanase, and β-glucosidase were increased by 1.1-fold to 4.4-fold in the *stk-12* mutant (Fig 2B–2D). These results suggested that the Δ*stk-12* mutant responded rapidly to cellulose to significantly increase cellulase production, and showed that STK-12 functions as a repressor of cellulase genes. Next, we compared the germination rates of WT and Δ*stk-12* on Avicel medium. The germination rate was significantly higher in Δ*stk-12* than in WT. At 6 hours post inoculation, 23.1% ± 4.9% of Δ*stk-12* conidia had germinated, compared with only 6.1% ± 2.2% of WT conidia (S2 Fig), indicating that Δ*stk-12* had a shortened lag phase when exposed to Avicel. This might, at least in part, contribute to faster cellulase production. Furthermore, we speculated that the *stk-12* mutant might accumulate more biomass. However, the results revealed that the dry weight of Δ*stk-12* was similar to that of the WT during incubation (S3 Fig). Thus, the hyper-secretion phenotype of this mutant was not due to an increase in biomass.

## Comparative analysis of transcriptomes between WT and Δ*stk-12* mutant

To determine the extent of the transcriptional changes conferred by *stk-12* disruption, we determined dynamic changes in the expressions of cellulase genes during batch culturing of Δ*stk-12* by RNA-seq. Biological replicate samples from the same time point clustered tightly in a principal component analysis (PCA) (S4 Fig), confirming that the RNA-seq data were highly reproducible. On the PCA plot, data from early time points (12 h and 24 h) clustered together, while data from later time points (72 h and 120 h) formed another cluster, indicating that the most significant transcriptional changes might occur at the later time points. As expected, the transcript levels of major cellulase genes showed no significant difference between WT and Δ*stk-12* in the first 24 h (Fig 3A, S2 Table). Of the 301 CAZy genes, only four glycoside hydrolase genes (NCU01517, NCU04554, NCU01080, and NCU07253) showed significantly altered transcript levels in Δ*stk-12*. These results indicated that STK-12 was not involved in the regulation of cellulase transcription during the initial period of growth on Avicel.

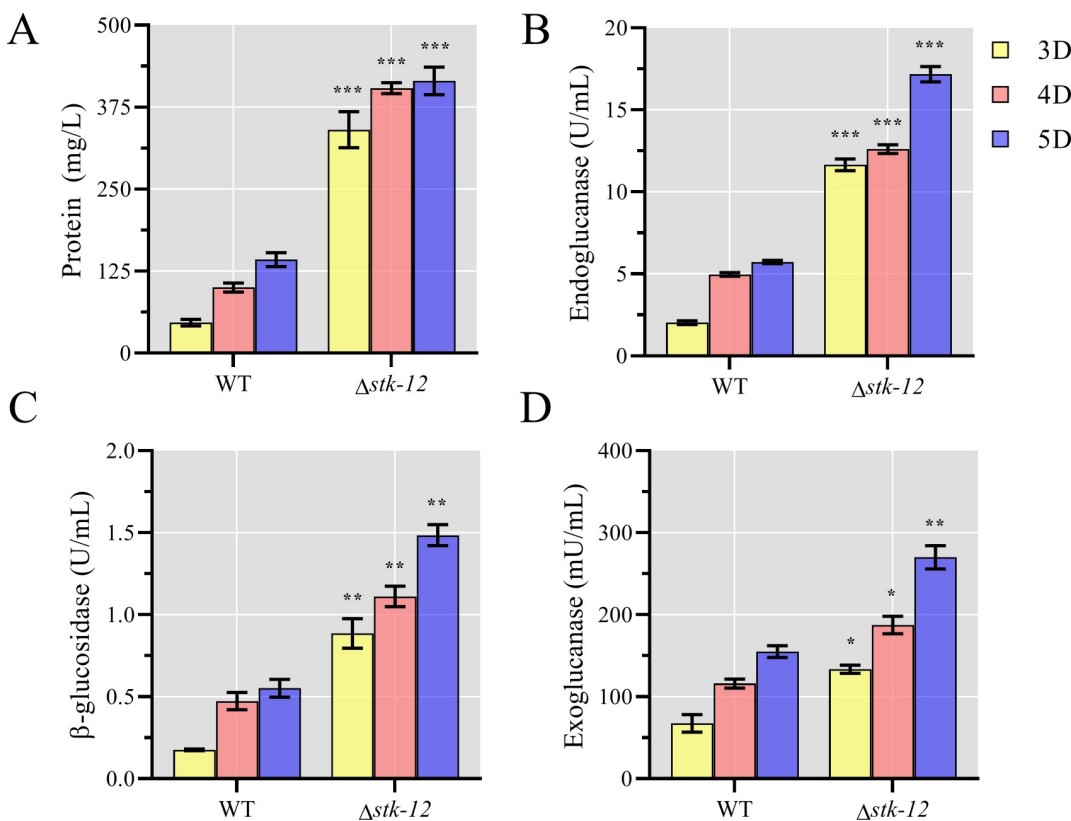

**Fig 2. Hyper-production of lignocellulases by *Neurospora crassa* due to deletion of *stk-12*.** Conidia of Δ*stk-12* and wild-type (WT) strains were separately inoculated into Avicel medium and batch cultured for 5 days. Total extracellular protein concentration (A), endoglucanase activity (B), β-glucosidase activity (C) and exoglucanase activity (D) were recorded. Values represent means of at least three replicates, error bars show standard deviation. Statistical significance was determined by two-tailed Student's *t*-test (*, $P < 0.05$; **, $P < 0.01$; ***, $P < 0.001$).

As mentioned above, the most significant changes in expression were observed after 3 days of culture (Fig 1I, S5 Fig). Analyses of the RNA-seq data revealed that the transcript levels of 528 genes were significantly higher and those of 294 genes were significantly lower in the Δ*stk-12* mutant than in WT in Avicel medium (S3 Table). Consistent with the cellulase activity data, of the 212 genes in the published Avicel regulon [10], 69 were significantly (more than two-fold) up-regulated in the Δ*stk-12* mutant. A FunCat analysis of the genes with elevated expression showed that 42 of them belonged to the "polysaccharide metabolism" group ($P = 1.36e^{-17}$). For example, the transcript levels of NCU07898, NCU00836, and NCU08760 encoding lytic polysaccharide monooxygenases (LPMOs) and NCU07340 (*cbh-1*), NCU09680 (*cbh-2*), and NCU04952 (*gh3-4*) encoding glycosyl hydrolases were significantly increased in the Δ*stk-12* strain compared with the WT (Fig 4A), consistent with the results shown in Fig 1I. The expression levels of NCU05923 and NCU00206, encoding cellobiose dehydrogenases, were increased more than 2.9- ($P = 0.0031$) and 5.5-fold ($P = 0.0073$), respectively, in the Δ*stk-12* strain compared with the WT (S3 Table). In addition to these CAZy genes, the essential cellulolytic regulator gene *clr-2* was up-regulated 2.2-fold ($P = 0.0093$), while the transcription levels of *clr-1* and *xyr-1* was not significantly changed. The transcript abundances of *res-1* and *rca-1* (encoding cellulase repressors) were decreased by 2.6- ($P = 0.0007$) and 5.8-fold ($P = 0.0002$), respectively (Fig 4B).

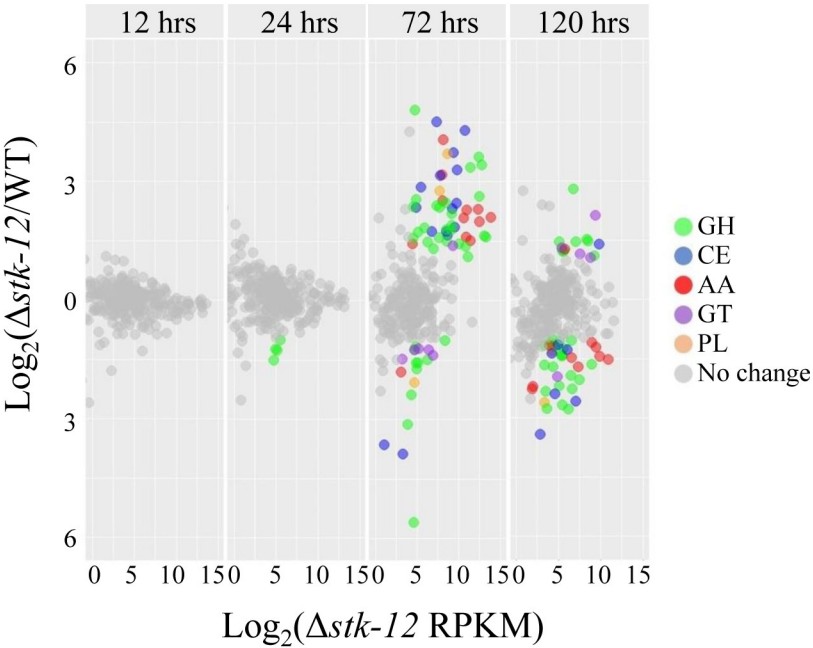

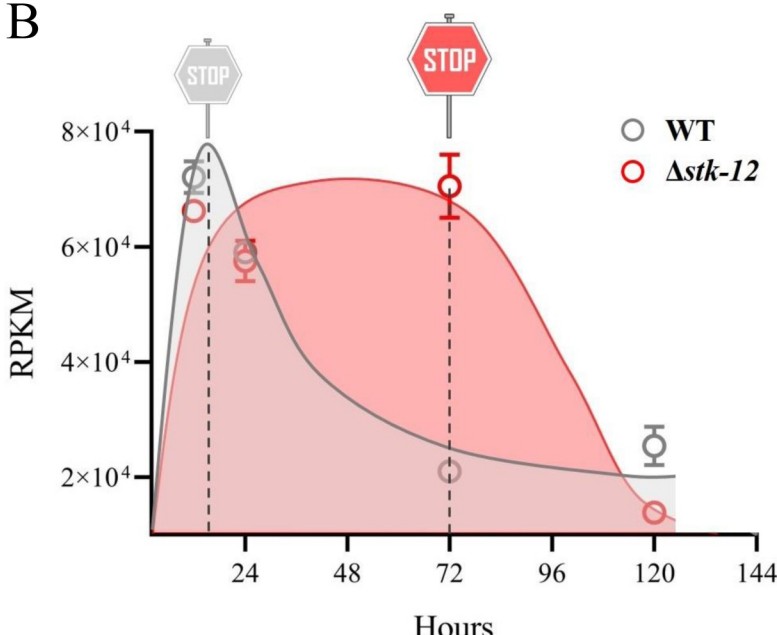

**Fig 3. Transcriptional profiling revealing roles of *stk-12* during growth on Avicel.** (A) Relative mRNA abundance of all CAZy genes in Δ*stk-12* vs. WT grown in Avicel medium. Genes with altered expression are plotted using different colors. Genes encoding glycoside hydrolases (GH, *green*), carbohydrate esterases (CE, *blue*), lytic polysaccharide monooxygenases (AA, *red*), glycosyl transferases (GT, *purple*), and polysaccharide lyases (PL, *orange*). Gray points represent genes with no differences in transcript levels between WT and Δ*stk-12*. (B) Expression levels of genes encoding cellulases in WT and Δ*stk-12* mutant during culture, as determined from RNA-seq data.

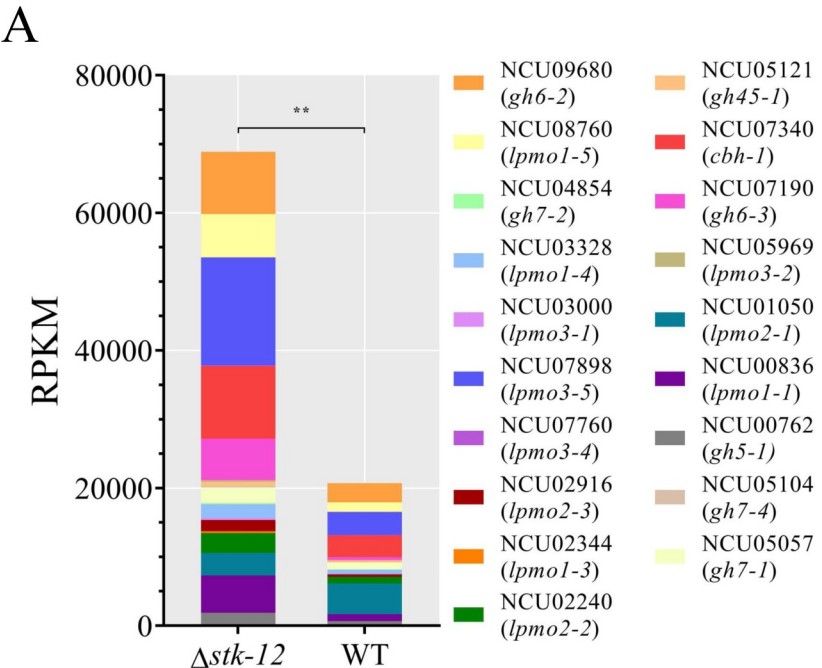

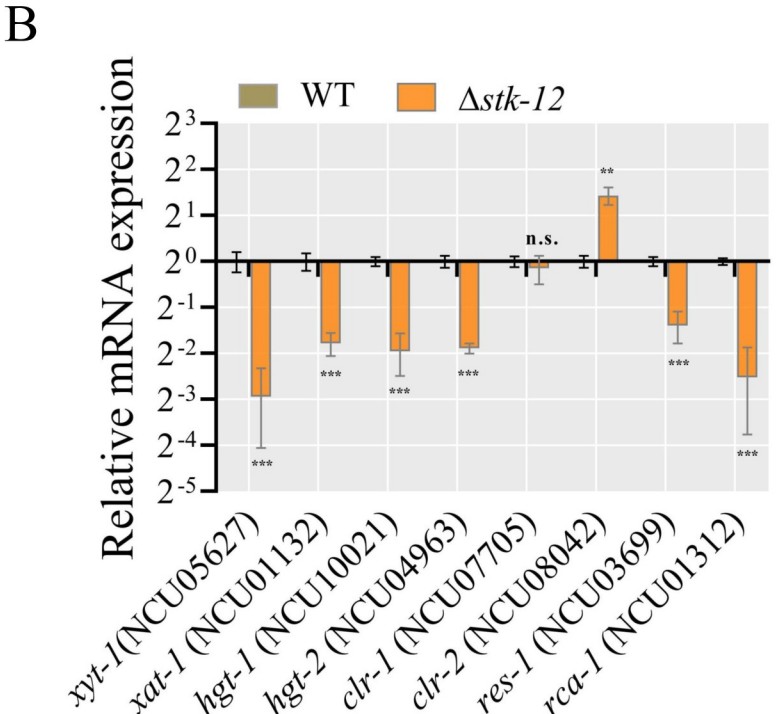

**Fig 4. Comparative analysis of gene transcription between Δstk-12 and WT grown on Avicel medium.** (A) Transcript levels (from RNA-seq data) of genes encoding cellulases in WT and Δstk-12 when grown on Avicel for 3 days. Values represent means of two biological replicates. (B) Transcript levels of genes encoding sugar transporters and transcription factors in Δstk-12 mutant relative to WT strain on Avicel. After Δstk-12 and WT conidia were grown on Avicel for 3 days, the transcript abundance of indicated genes was evaluated by quantitative real-time PCR. Values represent means of at least three biological replicates, error bars show standard deviation. Statistical significance was determined by two-tailed Student's $t$-test (**, $P < 0.01$; ***, $P < 0.001$, n. s., not significant).

Notably, the transcript levels of genes encoding several sugar transporters were significantly altered in Δ*stk-12*. In eukaryotes, sugar transporters are involved in nutrient uptake and in sensing of the external environment [7,44]. The genome of *N. crassa* contains 39 genes encoding putative sugar transporters [45]. Of the 39 candidate genes, 13 (33%) showed altered expression in the Δ*stk-12* strain compared with WT (Fig 4B and S3 Table). Nine of them showed dramatically increased expression (2-fold) in the Δ*stk-12* strain compared with WT. Two of them, *hgt-1* and *hgt-2*, encoding high-affinity glucose transporters, were found to be strongly up-regulated under glucose-limited conditions in WT *N. crassa* [44], but were down-regulated 3.8- (P = 0.0002) and 3.1-fold (P = 0.0001) in the *stk-12* mutant, respectively, suggesting an abnormal response to nutrient-limited conditions. Moreover, two high-affinity pentose transporter genes, *xat-1* and *xyt-1*, which have been shown to be significantly induced under L-arabinose or xylose conditions, respectively [46], were markedly down-regulated 3.4-(P = 0.0009) and 7.7-fold (P = 0.0008) in Δ*stk-12*, respectively (Fig 4B). These results suggest that the high-affinity nutrient-uptake system is compromised in the Δ*stk-12* mutant, and implies that nutrient response pathways play crucial roles in regulating cellulase gene expression.

The transcript levels of genes involved in the unfolded protein response (UPR) were also altered in the *stk-12* mutant (S3 Table). For example, NCU08897, encoding the translocation protein SEC61-1, was up-regulated 2.1-fold (P = 0.0006) in Δ*stk-12*. Also the transcript levels of NCU03213 and NCU01794, which encode components of the protein secretory pathway [47], were up-regulated by 5.9- (P = 0.0023) and 2.4-fold (P = 0.0099) in Δ*stk-12*, respectively. Furthermore, the transcript levels of some genes involved in the protein degradation pathway (NCU05852, NCU02636, and NCU05980) were significantly down-regulated by 4.2-(P = 0.0023), 2.7- (P = 0.0031) and 54-fold (P = 0.0003) in the Δ*stk-12* mutant than in the WT. Taken together, these results imply that the observed maintenance of high cellulase gene transcript levels in Δ*stk-12* leads to mild endoplasmic reticulum (ER) stress.

At the later period of growth in batch culture, the transcript levels of these CAZy genes were rapidly reduced to control levels or below the control levels. As shown in Fig 3A and S3 Table, after 5 days of growth on Avicel, there were no obvious differences in the transcript levels of genes encoding two major cellulases (*cbh-1* and *gh6-3*) between the Δ*stk-12* mutant and WT. However, another 40 CAZy genes were significantly down-regulated in Δ*stk-12*, including NCU05104 (encoding an exoglucanase), NCU05057, NCU05121, and NCU00762 (encoding endoglucanases), NCU00206 (encoding cellobiose dehydrogenase), and four LPMO genes (NCU01050, NCU03000, NCU08760 and NCU03328). Although the transcript levels of *clr-1* and *clr-2*, encoding two essential cellulase transcription factors, were not altered at the later stage (120 h), *cre-1*, encoding the major regulator of CCR, was significantly up-regulated (S3 Table), which might partly explain changes in the transcript levels of CAZy genes.

Overall, the transcriptomic analyses revealed that cellulase gene expression levels were maintained at high levels for 56 h in the Δ*stk-12* mutant, compared to only 8 h in the WT strain (Fig 3B). These results implied that deletion of *stk-12* increased the time required to activate RESS. Furthermore, our results demonstrated the mRNA stability of cellulase genes was not affected by the *stk-12* deletion (S6 Fig). Thus, increased cellulase production is therefore most likely due to long-term maintenance of high cellulase gene expression levels, which more likely caused by lacking of a brake to turn down the cellulase expression at transcriptional level.

## Expression level of *stk-12* affects cellulase production

As described above, deletion of *stk-12* positively affected lignocellulase production. Therefore, we hypothesized that excessive phosphorylation by STK-12 might inhibit cellulase production

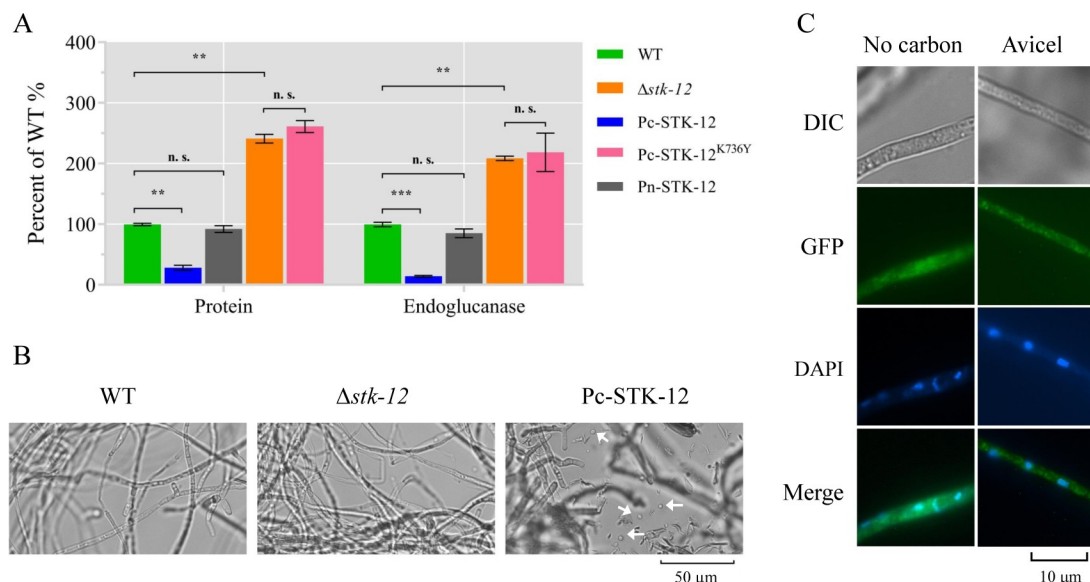

**Fig 5. Phenotype of Δ*stk-12* deletion mutant, complementation strain (Pn-STK-12), and mis-expression strains (Pc-STK-12 and Pc-STK-12^K736Y) when grown on Avicel medium.** (A) Total extracellular protein concentration and endoglucanase activity of culture broth were measured and are expressed as a percentage of those in WT. Values represent means of at least three biological replicates, error bars show standard deviation. Statistical significance was determined by two-tailed Student's *t*-test (\*\*, $P < 0.01$; \*\*\*, $P < 0.001$; n.s., not significant). (B) Mycelial morphologies of wild-type (WT) strain, Δ*stk-12* deletion mutant, and mis-expression strain Pc-STK-12 after 5 days of culture. Arrows indicate spores. Images were acquired under a OLYMPUS BX51 microscope using a QImaging Retiga 2000R camera. (C) Subcellular localization of STK-12 in *Neurospora crassa*. Strain with *stk-12* under control of *ccg-1* promoter was pre-grown in liquid MM with 2% (w/v) sucrose as sole carbon source for 16 h and then transferred into Vogel's salts with or without 2% (w/v) Avicel and grown for an additional 4 h. Location of STK-12 was monitored by GFP fluorescence. Nuclei were stained by DAPI.

in *N. crassa*. To test this hypothesis, we engineered Δ*stk-12* mutants expressing the GFP-STK-12 fusion protein under the control of either the native *stk-12* promoter (Pn-STK-12, complementation strain) or the clock controlled gene-1 (*ccg-1*) promoter (Pc-STK-12, overexpression strain). The level of extracellular protein produced by the complemented strain was reduced to that of the WT control (Fig 5A), corroborating that the hyper-secretion phenotype was due to the lack of *stk-12* rather than other unknown genetic mutations. As expected, *stk-12* expression levels in Pc-STK-12 strain were significantly higher than WT when grown on Avicel (11-fold) (S7 Fig). And the overexpression *stk-12* strain showed a consistent decrease in cellulase production and cellulolytic activity (Fig 5A). Surprisingly, however, overexpression of *stk-12* led to inappropriate conidiation in liquid culture (Fig 5B), while neither WT nor Δ*stk-12* exhibited submerged-culture conidiation. This raised the possibility that STK-12 has additional roles in development connected to nutrient sensing or nutrient responses.

A sequence alignment analysis suggested that K736 of STK-12 was likely to be necessary for its kinase activity [42]. To confirm that activation of STK-12 was required for proper cellulase regulation, we mutated Lys736 to Tyr (STK-12^K736Y) to mimic an inactive form of STK-12 and expressed this mutant protein in the Δ*stk-12* mutant. The inactive-form STK-12^K736Y was stably expressed as determined by RT-qPCR analyses (S7 Fig). However, it could not restore the WT phenotype (Fig 5A), confirming that this catalytic domain is required for its function.

## STK-12 is localized in the cytoplasm in *N. crassa*

To assess the subcellular localization of STK-12 protein, we tracked the C-terminal GFP-tagged STK-12 in the Δ*stk-12* strain. Under its native promoter, GFP-tagged STK-12

fluorescence was too weak to observe (S8 Fig), suggesting that the expression level of *stk-12* was very low, which was consistent with the RNA-seq data (S3 Table). To determine the subcellular localization of the STK-12 protein, we recorded the GFP signal of the STK-12-GFP fusion protein in young hyphae of the over-expression strain Pc-STK-12. After pre-growth in minimal medium for 16 h followed by culture in Avicel medium for another 4 h, STK-12-GFP was detected predominantly in the cytoplasm (Fig 5C) and also starvation did not cause a change in the cytoplasmic localization. Previous studies have shown that *S. cerevisiae* Rim15, the homolog of *N. crassa* STK-12, can translocate to the nucleus under glucose-limited conditions [35]. However, a similar phenotype was not observed in *N. crassa* (Fig 5C), suggesting that the regulation of STK-12 might function differently in *N. crassa*.

## IGO-1 is a direct downstream target of STK-12

To elucidate how STK-12 affects lignocellulase gene transcription, we tried to identify its crucial downstream targets. Phylogenetic analyses demonstrated that STK-12 is highly conserved in eukaryotes and is homologous to *S. cerevisiae* Rim15 (S9 Fig). Recent studies showed that Igo1 and Igo2 are direct targets of Rim15 [42,48]. The loss of both Igo1 and Igo2 appeared to largely phenocopy of the loss of Rim15 [42]. We therefore wondered whether the IGO homologs of *S. cerevisiae* in *N. crassa* could be downstream effectors also in the STK-12 pathway. Sequence alignment analyses revealed two *igo* orthologs (NCU03860 and NCU02932) in *N. crassa*; these two genes were designated *igo-1* and *igo-2*, respectively. We evaluated the cellulase production of the respective deletion strains through batch culturing with Avicel. The extracellular protein concentration of the Δ*igo-1* strain was increased by approximately 1.9-fold, and endoglucanase activity was increased by about 1.8-fold compared with the WT (Fig 6A), while the loss of *igo-2* did not affect cellulase production (S10 Fig). To further demonstrate that this phenotype was due to the loss of *igo-1*, we engineered the Δ*igo-1* strain expressing C-terminal GFP-tagged IGO-1 under the control of either the native promoter (complemented strain, Pn-IGO-1) or the constitutive *ccg-1* promoter (mis-expression strain, Pc-IGO-1). The results showed that the *igo-1* expression levels in Pc-IGO-1 strain were significantly higher than WT when grown on Avicel (4.5-fold) in *N. crassa* (S7 Fig). When these strains were batch cultured in Avicel medium, the complemented strain showed a secretion phenotype similar to that of the WT, while the Pc-IGO-1 strain exhibited significantly lower secreted protein levels and lower endoglucanase activity (Fig 6A). Fluorescence microscopy observations indicated that IGO-1 was uniformly distributed throughout the cytoplasm (Fig 6B).

The transcript level of *igo-1* was increased by 2.1-fold and 5.8-fold under Avicel and no-carbon conditions, respectively (S1 Fig). Intriguingly, the transcript abundance of *stk-12* was also significantly induced 3.6- to 8.7-fold by a 4-h exposure to nutrient-limiting conditions, indicating that these two genes might be co-expressed. The RT-qPCR results showed that the transcript levels of genes encoding major cellulases were significantly elevated in the Δ*igo-1* mutant compared with WT when grown on Avicel medium for 3 days, while the transcript levels of genes associated with high-affinity sugar transporters were lower (S11 Fig). Since similar transcriptional patterns were observed in the Δ*stk-12* mutant, the absence of *igo-1* appeared to largely phenocopy the loss of *stk-12*.

To test the interaction between STK-12 and IGO-1 *in vitro*, we conducted yeast two-hybrid assays by co-transforming AD-IGO-1 and BD-STK-12 into the yeast strain AH109. In these assays, IGO-1 interacted with STK-12 (Fig 6C). To test whether STK-12 binds to IGO-1 *in vivo*, we generated an STK-12-specific antibody. The antibody recognized a specific band at the predicted molecular weight in the WT and mis-expression strains, but not in the Δ*stk-12* mutant (Fig 6D). Co-immunoprecipitation (Co-IP) assays were performed using proteins

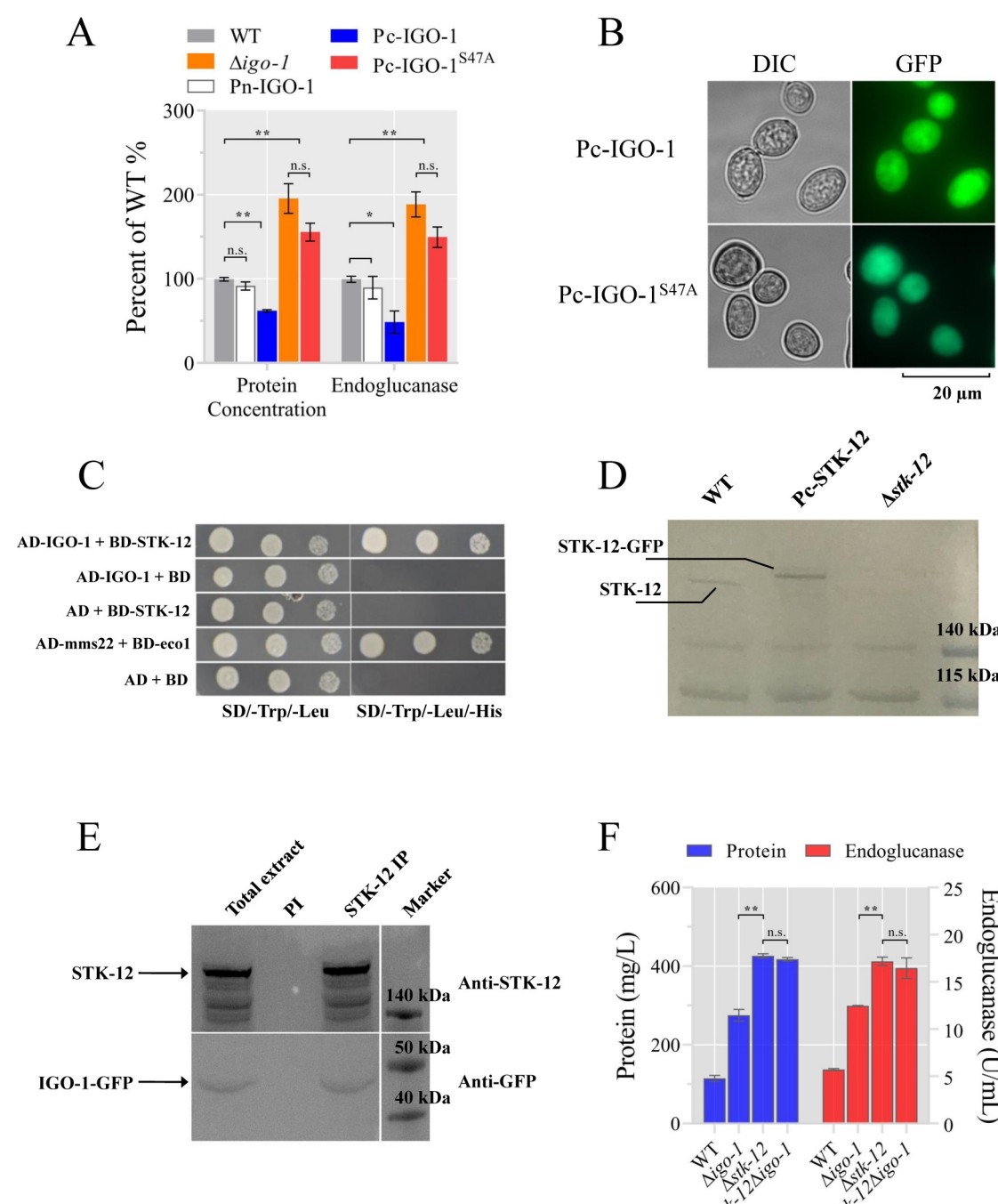

**Fig 6. STK-12 interacts with IGO-1. (A) Phenotype of wild-type (WT), Δ*igo-1*, Pc-NcIGO-1, and Pc-NcIGO-1^S47A strains when grown on Avicel medium.** Conidia from indicated strains were separately inoculated into Avicel medium and batch cultured for 5 days. Total extracellular protein concentration and endoglucanase activity of culture broth were measured and expressed as a percentage of those in WT. Values represent means of at least three biological replicates, error bars show standard deviation. Statistical significance was determined by two-tailed Student's *t*-test (*, $P < 0.05$; **, $P < 0.01$; n. s., not significant). (B) Subcellular localization of IGO-1 and its mutant derivative. Pc-NcIGO-1 and Pc-NcIGO-1^S47A strains were grown on MM plates for 5 days. (C) Yeast two-hybrid assay demonstrating interaction between STK-12 and NcIGO-1. Yeast cells were grown on SD/-Trp/-Leu or SD/-Trp/-Leu/-His medium for 3 days. (D) Western blot analyses showing that polyclonal antibody specifically recognizes STK-12 protein in WT and Pc-STK-12 strains but not in Δ*stk-12* mutant. Arrows indicate STK-12 or STK-12-GFP protein band detected by STK-12 polyclonal antibody. (E) Immunoprecipitation assays using STK-12 antiserum showing that STK-12 interacts with IGO-1 *in vivo*. PI, preimmune serum. (F) *stk-12* and *igo-1* probably act in same pathway to regulate cellulases. Conidia from WT and *stk-12* and *igo-1* single (Δ*stk-12*, Δ*igo-1*) and double (Δ*stk-12*Δ*igo-1*) mutants were inoculated into Avicel medium and batch-cultured

for 5 days. Total extracellular protein concentration and cellulase activity were measured. Values represent the means of at least three biological replicates, error bars show standard deviation. Statistical significance was determined by two-tailed Student's *t*-test (**, $P < 0.01$; n. s., not significant).

extracted from a strain expressing IGO-1-GFP. The results showed that IGO-1-GFP co-immunoprecipitated with STK-12 (Fig 6E). These results suggest that IGO-1 is the downstream target of STK-12, consistent with previous findings in *S. cerevisiae* [42,48].

To assess the possible genetic interactions between the genes encoding STK-12 and IGO-1, the Δ*stk-12*Δ*igo-1* double mutant was generated through crosses and its cellulase phenotype was determined. If STK-12 and IGO-1 operate through the same pathway, the cellulase phenotype should be similar in the Δ*stk-12* mutant and the Δ*stk-12*Δ*igo-1* mutant. As expected, after 5 days of culture in Avicel medium, the level of extracellular protein and cellulase activity in the Δ*stk-12*Δ*igo-1* mutant were indistinguishable from those of the Δ*stk-12* strain (Fig 6F), indicating that Δ*stk-12* was epistatic to Δ*igo-1*. Taken together, these data suggest that the STK-12 pathway acts through IGO-1 to regulate cellulase gene expression in *N. crassa*.

## IGO-1 is activated by STK-12 via phosphorylation

To further understand the molecular function of IGO-1, we searched for possible phosphorylation sites in its protein sequence according to previously published data [42,49]. Considering these data, S47 of *N. crassa* IGO-1 was determined to be a likely target for phosphorylation by STK-12. This amino acid is highly conserved among homologs of IGO-1 in cellulolytic fungi (Fig 7A). To assess the possible function of this potential STK-12 phosphorylation site, an inactive form of *N. crassa* IGO-1 was created (IGO-1$^{S47A}$) and was transformed into the *his-3;Δigo-1* strain at the *his-3* locus (strain Pc-IGO-1$^{S47A}$). After 5 days of culture, the protein secretion titer and endoglucanase activity of Pc-IGO-1$^{S47A}$ were found to be increased by more than 60% and 54%, respectively, compared to the WT (Fig 6A). While the phosphor-site did not appear to influence its expression and subcellular localization (Fig 6B, S7 Fig), the results indicate that phosphorylation at S47 of IGO-1 is required for its function in cellulase regulation in *N. crassa*.

To further verify that IGO-1 is phosphorylated by STK-12 at S47, WT *N. crassa* IGO-1 and the phosphorylated active form (IGO-1$^{S47D}$) were each transformed into the *his-3;Δigo-1Δstk-12* strain. As expected, constitutive expression of IGO-1 in Δ*igo-1*Δ*stk-12* could not repress the hyperproduction phenotype due to the lack of activation by STK-12. However, the cellulase phenotype in the Δ*stk-12*Δ*igo-1* mutant was attenuated by expression of IGO-1$^{S47D}$ (Fig 7B), suggesting that Ser47 phosphorylation mediated by STK-12 is essential for the function of IGO-1. Furthermore, the IGO-1 and its mutants are stably expressed as determined by RT-qPCR analysis (S7 Fig). These results indicate that phosphorylation by STK-12 is essential for IGO-1 activation in *N. crassa*.

## Effect of protein phosphatase 2A on cellulase production

Previous studies have shown that RIM15 phosphorylates IGO1/2 to inhibit the activity of protein phosphatase 2A (PP2A) [50], a serine/threonine phosphatase that plays diverse roles in transcription, translation, and signal transduction [51,52]. More recently, seven phosphatases whose mutants exhibited severe defects in protein production and cellulolytic enzyme activities were shown be involved in cellulase production in *A. nidulans* [28]. To check whether PP2A is involved in the regulation of cellulase production in *N. crassa*, we tested the cellulase production capacity of the PP2A mutant. The result showed that the deletion of the subunit of PP2A encoded by NCU06563 led to a hypo-production phenotype (Fig 8A). Furthermore, fluorescence microscopy observations indicated that PP2A was uniformly distributed

A

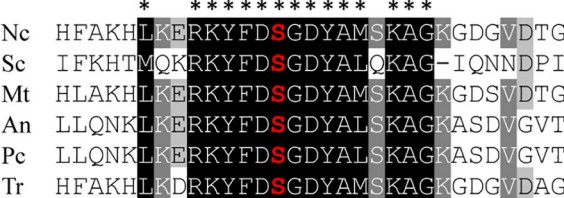

```
          *   * * * * * * * * * *   * * *
Nc  HFAKHLKERKYFDSGDYAMSKAGKGDGVDTG
Sc  IFKHTMQKRKYFDSGDYALQKAG-IQNNDPI
Mt  HLAKHLKERKYFDSGDYAMSKAGKGDSVDTG
An  LLQNKLKERKYFDSGDYALSKAGKASDVGVT
Pc  LLQNKLKERKYFDSGDYALSKAGKASDVGVT
Tr  HFAKHLKDRKYFDSGDYAMSKAGKGDGVDAG
```

B

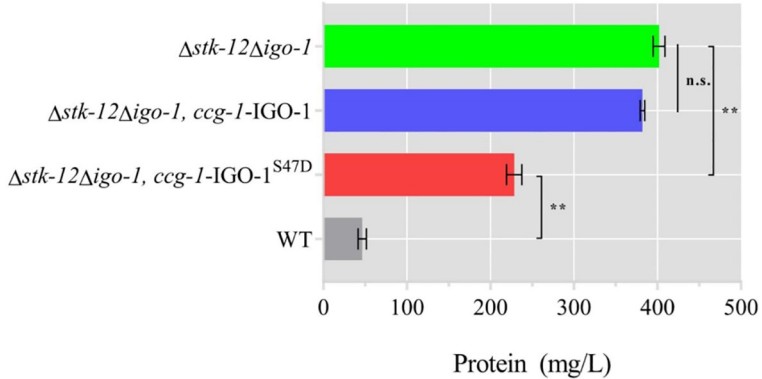

**Fig 7. IGO-1 is activated by STK-12 via phosphorylation.** (A) Amino acid sequence alignment of conserved domain from IGO-1 homologs in *Neurospora* (NCU03860), *Saccharomyces* (NP_014242.1), *Myceliophthora* (XP_003664474.1), *Aspergillus* (XP_001392163.1), *Penicillium* (EPS31852.1) and *Trichoderma* (XP_006963064.1). Asterisks indicate conserved amino acids; phosphorylation site is shown in red. (B) Mutational analysis of S47 of IGO-1. *N. crassa* WT strain, Δ*stk-12*Δ*igo-1*, and mutants expressing either a wild-type copy of *igo-1* or mutated *igo-1* allele were separately inoculated into Avicel medium and batch-cultured for 3 days. Total extracellular protein concentration in culture broth was measured. Values represent means of at least three biological replicates, error bars show standard deviation. Statistical significance was determined by two-tailed Student's *t*-test (**, $P < 0.01$; n. s., not significant).

throughout the cytoplasm (S12 Fig). To further explore the roles of this gene in cellulase gene transcription, we examined the kinetics of cellulase gene expression in the ΔNCU06563 using RT-qPCR. Although deletion of NCU06563 led to severe defects in cellulase production, the expression level of major cellulase gene (*cbh-1*) was not affected during the initial period of batch culture (24 h) (Fig 8B). It is striking that the transcript abundance of *cbh-1* gene was found to be dramatically down-regulated (33-fold) in ΔNCU06563 compared to WT when grown on Avicel for 72 h (Fig 8B), suggesting that the expression of *cbh-1* declined more rapidly in the ΔNCU06563 mutant. As described in previous results, the major cellulase gene *cbh-1* expression level was maintained at a high level for up to 72 h in the Δ*stk-12* (Fig 1I, Fig 3B), while *cbh-1* gene exhibited a completely opposite expression pattern in the ΔNCU06563 mutant. Based on this evidence, we speculated that loss of *stk-12* led to a failure to activate IGO-1, thereby resulting in elevated PP2A activity. This may be one reason for the changes in cellulase production. However, further detailed experiments are needed to evaluate the roles of PP2A in regulating cellulase gene expression.

## Ultrastructural analysis of the Δ*stk-12* mutant

The ER plays essential roles in protein synthesis and folding. Previous studies have demonstrated that increased secretory activity is correlated with the proliferation of ER [53]. As

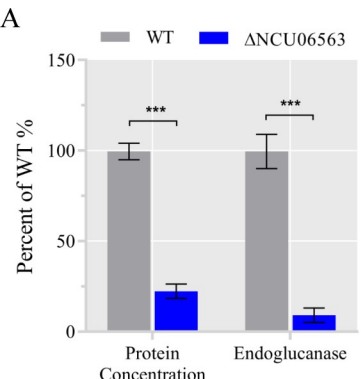
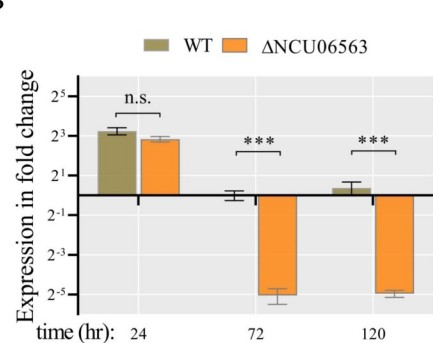

**Fig 8. Phenotype of WT and ΔNCU06563 strains when grown on Avicel medium.** Conidia from ΔNCU06563 and wild-type (WT) strains were separately inoculated into Avicel medium and batch-cultured for 5 days. (A) Deletion of NCU06563 results in serious defects in cellulase production. Total extracellular protein concentration and endoglucanase activity of culture broth were measured and are expressed as a percentage of those in WT. (B) Relative transcript levels of major cellulase gene (*cbh-1*) in hypo-producing mutant ΔNCU06563 versus WT after 1–5 days growth on Avicel. After growth of conidia in Avicel for 24, 72, or 120 h, transcript abundance of *cbh-1* was evaluated by RT-qPCR. Values represent means of at least three biological replicates, error bars show standard deviation. Statistical significance was determined by two-tailed Student's *t*-test (\*\*\*, $P < 0.001$). Data are normalized against tested gene's transcript level in WT strain at 72 h. *Actin* (NCU04173) transcript level was used as endogenous control in all samples. Values represent means of at least three biological replicates; error bars show standard deviation.

described above, the Δ*stk-12* mutant displayed a rapid response to cellulose and a high rate of protein biosynthesis during the initial growth period on Avicel (S5 Fig). We wondered whether these phenotypes might be related to observable differences in ER development potentially mediated by STK-12. To test this hypothesis, we analyzed the ultrastructure of the Δ*stk-12* mutant by transmission electron microscopy (TEM). However, despite a 7-fold higher cellulase production than that of the WT after 3 days (Fig 2A and S5 Fig), the Δ*stk-12* mutant did not show a proliferation of rough ER when grown on Avicel medium for the same time period (S13 Fig). This appeared to contrast with the results of previous studies [53,54]. However, several ultrastructural changes were observed in the Δ*stk-12* mutant hyphae during the initial period of batch culture. When grown in Avicel medium for 24 h, the Δ*stk-12* mutant hyphae contained large amounts of ER (Fig 9A and 9C), whilst comparably little ER was present in the WT hyphae at the same time (Fig 9B and 9D). This change might be one reason for the higher production of cellulases in Δ*stk-12*. Surprisingly, this phenomenon disappeared as fermentation progressed. After 48 h of growth in Avicel medium, mycelia from the WT strain and the Δ*stk-12* mutant contained similar amounts of ER (Fig 9E–9H). The existence of ultrastructural differences at the initial stage of fermentation implied that these distinct features resulted from a genetic modification rather than a response to increased protein secretion, suggesting that deletion of *stk-12* might accelerate cell development. In addition to ER, the Δ*stk-12* mutant contained abundant mitochondria when grown on Avicel for 48 h (Fig 9G), implying that Δ*stk-12* cells might increase the number of mitochondria to supply more energy for protein synthesis.

## Genetic interaction between STK-12 and STK-10

In our screening analyses, the deletion of *stk-10* (NCU03200), a homolog of *S. cerevisiae sch9*, led to severe defects in cellulase production (Fig 1). A previous study demonstrated that Sch9 directly inhibits Rim15 function by phosphorylation under carbon-limited conditions [49], indicating that STK-10 might function in a similar manner to regulate STK-12. To further explore the genetic interaction between STK-10 and STK-12, we generated the Δ*stk-10*Δ*stk-12*

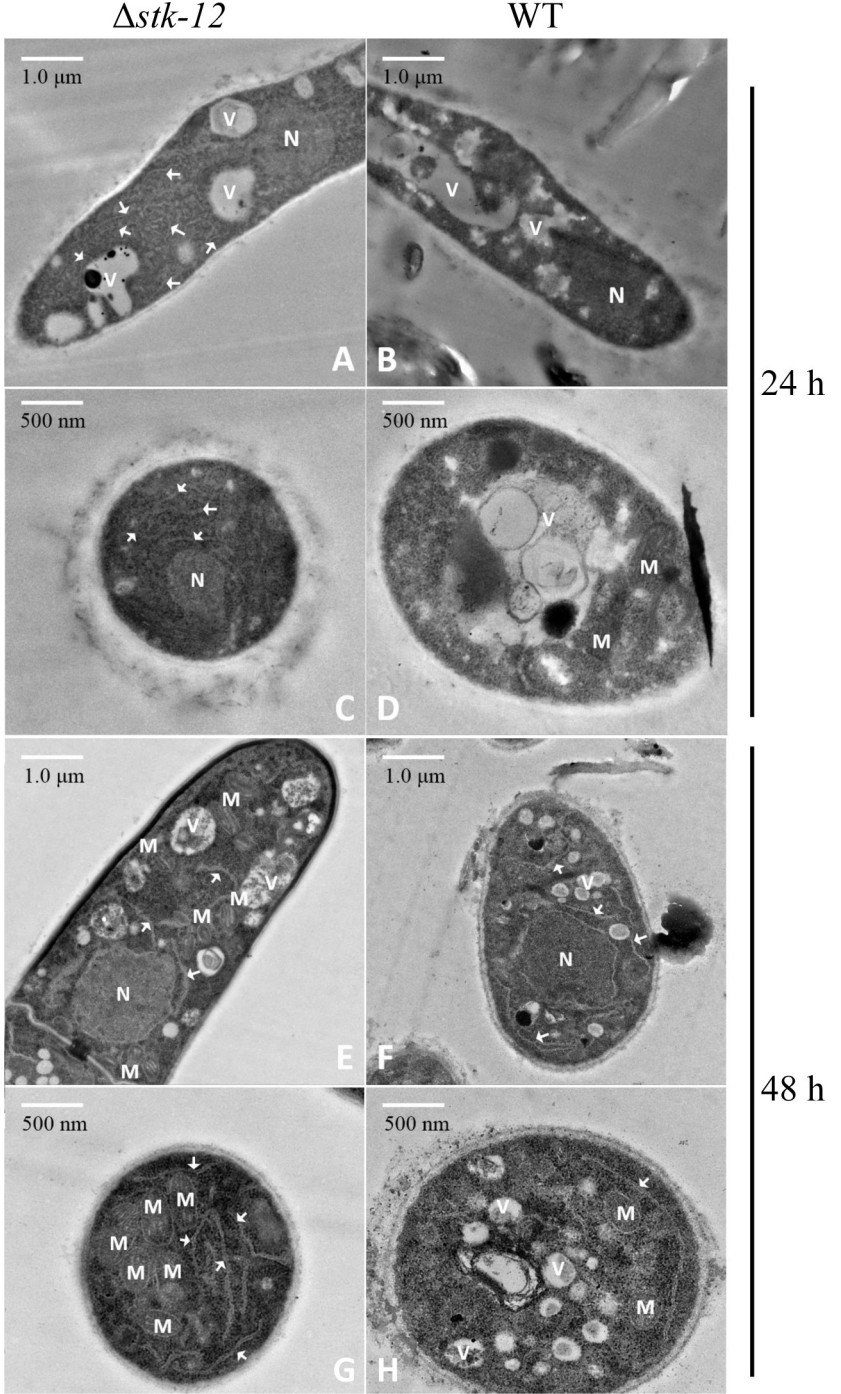

**Fig 9. Transmission electron micrographs of *N. crassa* wild-type (WT) strain and *stk-12* deletion strain.** Δ*stk-12* mutant (A, C, E and G) and *N. crassa* WT strain (B, D, F and H) were grown in minimal medium with 2% (w/v) Avicel as sole carbon source for 24 h (A-D) or 48 h (E-H). Mycelia were collected and prepared for transmission electron microscopy. White arrows indicate endoplasmic reticulum. M, mitochondrion; N, nucleus; V, vacuole.

double mutant by crossing. As predicted, protein production and endoglucanase activity of the Δ*stk-10*Δ*stk-12* strain were restored to WT levels after 5 days of growth on Avicel (Fig 10). Overexpression of STK-12 led to defects in cellulase expression (Fig 5A). Taken together, these

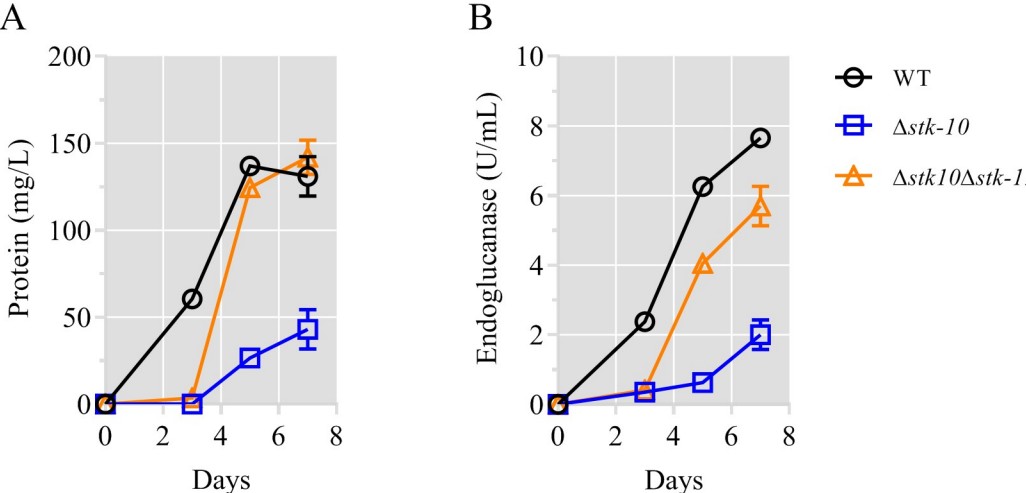

**Fig 10. Deletion of *stk-12* rescues phenotype of *stk-10* mutant on Avicel.** Total extracellular protein concentration (A) and endoglucanase activity (B) in culture broth of WT and mutants (Δ*stk-10* and Δ*stk-12*Δ*stk-10*) during growth in cellulose medium. Values represent means of at least three biological replicates, error bars show standard deviation.

findings support the model that STK-10 negatively regulates STK-12 and suggest that the failure to inhibit STK-12 activity is detrimental to cellulose utilization. Moreover, although deletion of *stk-12* in the Δ*stk-10* background restored utilization of Avicel, there was an obvious lag in cellulase production in the double mutant as compared with the WT (Fig 10). This result suggests that additional unknown downstream effectors of the STK-10 signaling cascade are involved in the regulation of cellulase expression.

## STK-12 pathway is independent of the *cre1*-mediated CCR pathway

In filamentous fungi, *cre-1*-mediated CCR is the main known mechanism of the suppression of cellulase gene expression. To check whether cross-talk exists between the STK-12 pathway and the CRE-1 pathway, a Δ*stk-12*Δ*cre-1* double mutant was generated by crossing. The abilities of Δ*stk-12*, Δ*cre-1*, and Δ*stk-12*Δ*cre-1* to secrete cellulases in Avicel medium were compared. The double mutant exhibited stronger cellulase production capability than either the Δ*stk-12* strain or the Δ*cre-1* strain. After 3 days of batch culture, the endoglucanase activity and secreted protein in the double deletion strain Δ*stk-12*Δ*cre-1* were 71% and 60% higher, respectively, than those of Δ*stk-12* (Fig 11A). Deleting both *stk-12* and *cre-1* had a pronounced additive effect on cellulase production, indicating that STK-12 and CRE-1 probably work through separate pathways to regulate cellulase gene expression.

We also evaluated the resistance of the Δ*stk-12* mutant to 2-deoxy-glucose (2-DG), a non-metabolizable D-glucose analog that is often used to evaluate the impairment of CCR in filamentous fungi [55]. Strains with defective CCR display 2-DG resistance. Consistent with previous observations, the Δ*cre-1* mutant showed 2-DG resistance when 2% Avicel and 0.1% 2-DG were used as carbon sources [55]. However, the Δ*stk-12* mutant and the WT strain were sensitive to 2-DG and could not grow in Avicel medium (Fig 11B), implicating that CCR is functional in the Δ*stk-12* mutant. These results suggest that STK-12 is not involved in CCR in *N. crassa*, but instead acts through a novel cellulase regulation pathway to regulate cellulolytic gene expression.

## STK-12 is widely distributed in filamentous fungi

As described above, inhibition of the STK-12-IGO-1 pathway led to significantly enhanced hydrolytic enzyme production and a shortened fermentation period. These characteristics are

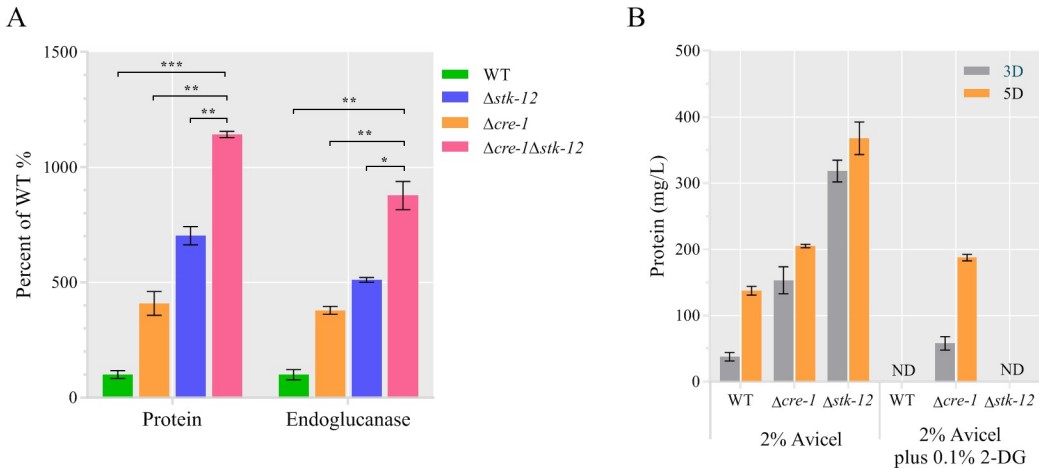

**Fig 11. Enhanced cellulase production in the Δ*stk-12* mutant does not result from relief from catabolite repression mediated by *cre-1*.** (A) Conidia from Δ*cre-1*Δ*stk-12*, Δ*cre-1*, Δ*stk-12*, and wild-type (WT) strains were separately inoculated into Avicel medium and batch-cultured for 3 days. Total extracellular protein concentration and endoglucanase activity of culture broth are expressed as percentages of those in WT. Values are means of at least three biological replicates, error bars show standard deviation. Statistical significance was determined by two-tailed Student's *t*-test (*, $P < 0.05$; **, $P < 0.01$; ***, $P < 0.001$). (B) Cellulase production in Avicel medium with or without 0.1% 2-deoxy-glucose (2-DG). Total extracellular protein concentration was measured after 5 days of culture. Values are means of at least three biological replicates, error bars show standard deviation. ND, not detected.

desirable for industrial applications, suggesting that this pathway may contain novel targets for the improvement of cellulase production in industrial strains. A phylogenetic analysis demonstrated that STK-12 is widely distributed in filamentous fungi (S9 Fig). To test the hypothesis that the role of STK-12 is conserved in other filamentous fungi, we created a deletion strain for its ortholog in the thermophilic cellulolytic fungus *Myceliophthora thermophila* (XP_0036597 08.1, named *Mtstk-12*). Similar to the *N. crassa* Δ*stk-12* mutant, the *M. thermophila* Δ*Mtstk-12* deletion strain showed a moderate increase in cellulase production (Fig 12), suggesting that the function of STK-12 in cellulolytic fungi may be conserved.

## Discussion

Protein kinases are well-known regulators of gene expression. A previous study illustrated that extensive post-translational regulation occurs in *N. crassa* when it is exposed to Avicel [49,56]. Screening for *N. crassa* hyper-producers of cellulases also showed that loss of the serine/threonine protein kinase gene *prk-6* increased the secreted protein levels [13]. These results indicated that protein kinases probably perform important roles in the regulation of cellulase gene expression. Thus, in this study, we systematically screened the 64 *N. crassa* serine/threonine protein kinase mutants to identify the key components involved in the regulation of cellulase gene expression. Several negative regulators whose mutants showed increased cellulase production were identified. For example, deletion of *prk-2* (NCU07872), which encodes a homolog of *S. cerevisiae yak1*, resulted in enhanced cellulase production, consistent with previous observations in *T. reesei* [27]. *mik-1* (NCU02234) encodes a MAPKK kinase that functions in the protein kinase C signaling pathway controlling cell wall integrity. The hyper-production phenotype induced by a defect in *mik-1* implied a close link between cell wall integrity and cellulase secretion. In support of this hypothesis, a *T. reesei* Δ*tmk2* mutant showed defects in cell wall integrity and increased cellulase expression [22]. In *S. cerevisiae*, SNF1 plays critical roles in the CCR pathway, and is required for the nuclear export of the CCR transcription factor

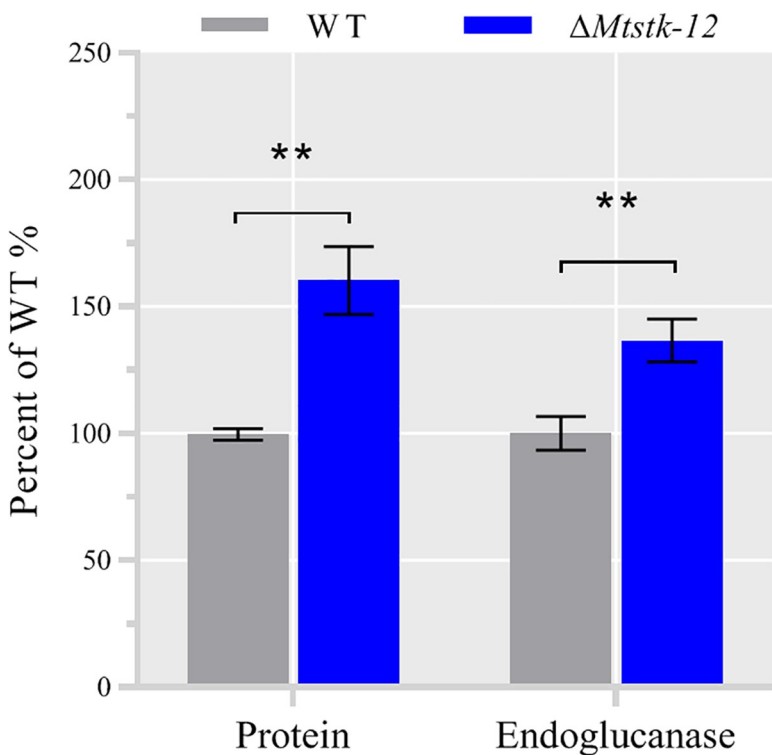

**Fig 12. The *M. thermophila Mtstk-12* deletion mutant shows a hyper-production phenotype under cellulolytic conditions.** Avicel broth cultures were inoculated with conidia and grown for 5 days. Total extracellular protein concentration and endoglucanase activity of culture broth were measured and are expressed as a percentage of those in WT. Values represent means of at least three biological replicates, error bars show standard deviation. Statistical significance was determined by two-tailed Student's t-test (**, $P < 0.01$).

MIG-1 (ortholog of CRE-1/CreA). The loss of *snfA* from *Aspergillus nidulans* resulted in mislocation of CreA to the nucleus, and defective cellulase production [22]. However, deletion of *prk-10* (NCU04566), a homolog of *S. cerevisiae snf1*, led to enhanced cellulase production, suggesting that PRK-10 functions in regulating cellulase gene expression in a very different manner. Consistent with our hypothesis, it has been shown in *T. reesei* that the phosphorylation of CRE1 is mediated by a casein kinase II-like protein, and not by the SNF1 homolog [57]. Moreover, a recent study also revealed that cellular localization of CRE-1 does not play a critical role in regulating CCR in *N. crassa* [16]. Thus, a possible role of PRK-10 in the regulation of cellulase biosynthesis warrants a more detailed investigation.

Previous work on *S. cerevisiae* demonstrated that TOS3 can activate SNF1 by phosphorylation [58]. Our results showed that deletion of *stk-22* or *prk-10*, the homologs of *S. cerevisiae tos3* and *snf1*, respectively, had a similar effect on cellulase production in *N. crassa*, implying that STK-22 and PRK-10 might function in the same pathway in regulating cellulase formation. However, this hypothesis remains to be tested experimentally.

Of most interest, we found that the loss of *stk-12* (NCU07378) in *N. crassa* not only led to a hyper-production phenotype but also greatly shortened the fermentation period (Fig 2, S5 Fig), suggesting that STK-12 acts as a critical repressor limiting cellulase gene expression. Moreover, in medium-shift experiments, the transcript level of *stk-12* was dramatically up-regulated under carbon-limited conditions (S1 Fig). The mis-expression strain of *stk-12* (Pc-STK-12) showed inappropriate conidiation when it was batch-cultured in Avicel medium (Fig 5B), implying that nutrient sensing might be compromised. These results indicate that the STK-12

kinase cascade might be a nutrient signal transduction pathway that responds to nutrient levels. Therefore, we speculated that STK-12 plays a critical role in the cellular responses to low nutrient levels by turning down nutrient-consuming processes such as growth, metabolism, and protein translation, which was strongly supported by our transcriptional analyses (Fig 3B and S4 Table) and microscope observations (S2 Fig and Fig 9).

In *S. cerevisiae*, RIM15, the homolog of *N. crassa* STK-12, is derepressed under nutrient-limited conditions, and it phosphorylates its downstream effector IGO. Then, the activated IGO participates in multiple pathways in response to nutrient deprivation. In our analyses, the *N. crassa* homolog IGO-1 was a direct target of STK-12, and its phosphorylation by STK-12 was required to execute its function. These results suggested that the STK-12-IGO-1 pathway is highly conserved between yeast and filamentous fungi. Like cellulolytic fungi, plant/insect pathogenic fungi also secrete hydrolytic enzymes that degrade components of the host epidermis during the infection process [59]. Thus, further analyses of the functions of STK-12 orthologs in plant/insect pathogenic fungi may aid in the development of novel anti-fungal strategies and more potent strains for insect biocontrol.

Our screening analyses also identified several deletion strains with severely defective cellulase production, suggesting that the corresponding genes probably play positive roles in regulating cellulase production. Among these regulators, STK-10, the homolog of *S. cerevisiae* Sch9, is a critical component of the TORC1 signaling pathway, which is important for cell metabolism and proliferation. Deletion of *stk-10* severely decreased cellulase production (Fig 1), implying that the TORC1 pathway plays an important role in regulating cellulases. In *A. nidulans*. loss of *schA*, the homolog of *N. crassa stk-10*, resulted in mis-localization of CreA to the nucleus, and caused defects in protein production and cellulolytic enzyme activities, indicating that *schA* is necessary for CreA de-repression under cellulosic conditions [28]. However, a recent study showed that *N. crassa* CRE-1 always localized to nuclei under either glucose or Avicel conditions, implying that STK-10 might be involved in regulating cellulase biosynthesis in a different manner. In our analyses, deletion of *stk-12* in the Δ*stk-10* background restored cellulase production to WT levels, indicating that STK-10 might, at least in part, exert its function on cellulase production via STK-12.

In summary, we systematically screened protein kinase mutants of *N. crassa* and identified several potential genes that affect lignocellulase production. According to our results, the serine/threonine protein kinase STK-12 functions as a novel repressor of cellulase gene expression. We also identified IGO-1 as the major downstream effector of STK-12 in the repression of cellulase production. Our analyses showed that STK-12 is involved in the transcriptional repression of cellulase genes and delays the development of the endomembrane system under lignocellulolytic conditions, suggesting that it plays a critical role in the response to low nutrient levels by turning down nutrient-consuming processes. More importantly, our data indicate that the expression of cellulase genes is integrating signals from several signal pathways, such as the TORC1 pathway, the CCR pathway, and the STK-12 pathway, for which a complex interplay seems to exist (Fig 13). These findings provide a deeper insight into the mechanisms underlying cellulase regulation and may facilitate the improvement of industrial strains for increased lignocellulase production.

## Materials and methods

### *N. crassa* strains

*N. crassa* strains were obtained from the Fungal Genetic Stock Centre (http://www.fgsc.net), and included the wild-type strain (WT, FGSC 2489, A), two *his-3* mutant strains (FGSC 6103, A and FGSC 9716, a), and a set of 64 serine-threonine kinase mutants [31]. The *his-3*;Δ*stk-12*

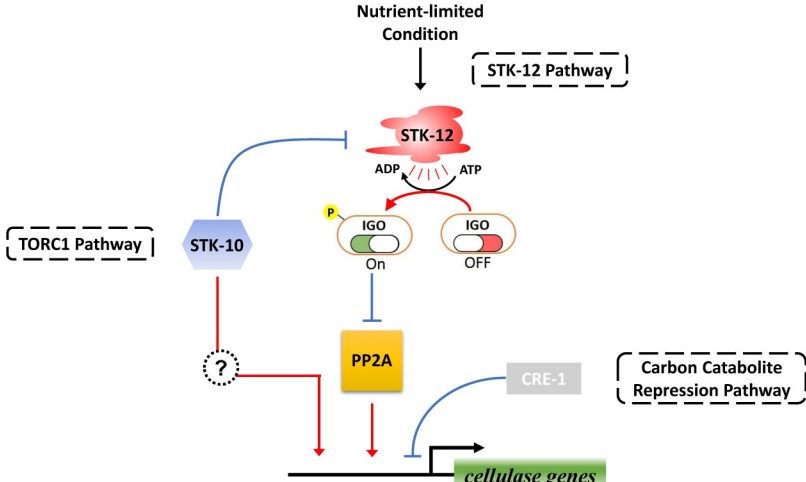

**Fig 13. Schematic model for the proposed roles of STK-12 in regulating cellulolytic gene expression in *N. crassa*.** Under nutrient-limited conditions, STK-12 is activated and subsequently triggers the activation of downstream effector IGO-1 by phosphorylation. Phosphorylated IGO-1 serves as a molecular switch to regulate cellulase gene expression by inhibition its downstream target, PP2A, which is necessary for utilization of cellulose in *N. crassa*. Furthermore, STK-10 is at least partly speculated to regulate cellulase gene expression via STK-12. And STK-12 and CRE-1 probably work through separate pathways to regulate cellulase gene expression.

strain, the *his-3;Δigo-1* strain, the *Δstk-12Δcre-1* strain, the *Δstk-12Δstk-10* strain, and the *his-3; Δstk-12Δigo-1* strain were created by performing sequential crosses as previously described (http://www.fgsc.net/Neurospora/NeurosporaProtocolGuide.htm). All constructed strains were confirmed by PCR with primer pairs (S5 Table). The mis-expression strains were constructed by transforming the corresponding strains with the linearized plasmid pMF272 harboring the promoter of the clock controlled gene 1 (*ccg-1*), various gene-coding sequences or point-mutated analogs, and regions flanking the *his-3* gene sequence.

## Culture conditions

To obtain conidia, *N. crassa* strains were propagated on slants of Vogel's minimal medium (VMM) [60] containing 2% w/v sucrose for 2 days in darkness at 28˚C, then in constant light for 8 days at room temperature. For crosses, the female parent was grown on plates of Westergaard medium [61] at room temperature until protoperithecia became visible. Conidia of the male parent were added to the plates for fertilization. After 3–4 weeks of cultivation at room temperature, ascospores were harvested and activated, plated on VMM supplemented with BDES, and incubated at 28˚C for 16 h [62]. Germinated ascospores were transferred to slants of 2% sucrose VMM and then screened by PCR. For liquid cultures, *N. crassa* conidia were inoculated into 100 mL 2% w/v Avicel VMM at $10^5$ conidia/mL and grown at 25˚C in constant light with shaking (200 rpm). For medium-shift experiments, conidia were inoculated at $10^5$ conidia/mL into 100 mL 2% w/v sucrose VMM and cultured for 16 h. Then, the cultures were centrifuged at 2,000 g for 10 min at 4˚C and washed three time with 1×Vogel's salts, then grown for 4 h in 100 mL VMM with 2% w/v carbon source (Avicel, glucose) or without a carbon source.

## Fungal biomass assay

The dry weight of mycelia was indirectly determined as previously described with some modifications [63]. Briefly, the mycelia from 10 mL of culture broth were harvested by

centrifugation at 4,000 g for 20 min at 4˚C and washed three times with deionized water. Then, the mycelia pellet was dried and weighed. The dried residue was resuspended in 5 mL of acetic nitric reagent (80:20, v/v), and bathed with boiling water for 2 hours to remove mycelia. The residual Avicel was washed in water three times, dried, and reweighed. Mycelial dry weight was defined as the dry weight of the original 10 mL culture minus that of residual Avicel.

## Plasmid construction and transformation

To rescue Δ*stk-12*, the open reading frame (ORF) of *stk-12* was cloned by PCR with the primers STK-12-ORF-F and STK-12-ORF-R. The resulting fragment was inserted into the *Xba*I and *Pac*I sites of pMF272 to form Pc-stk-12-gfp. The promoter of *stk-12* was amplified with the primers STK-12-PF and STK-12-PR, and then cloned into the *Not*I and *Xba*I sites of Pc-stk-12-gfp to generate Pn-stk-12-gfp.

To complement Δ*igo-1*, the full-length ORF of *igo-1* was cloned from WT cDNA using the primers IGO-ORF-F and IGO-ORF-R, digested with *Xba*I and *Pac*I, and then cloned into pMF272 to produce Pc-Ncigo-1-gfp. The promoter of *igo-1* was amplified with the primers IGO-PF and IGO-PR, and then cloned into the *Not*I and *Xba*I sites of Pc-Ncigo-1-gfp to generate Pn-Ncigo-1-gfp.

To overexpress NCU06563, the ORF was cloned from WT cDNA using the primers NCU06563-F and NCU06563-R, digested with *Xba*I and *Pac*I, and then cloned into pMF272 to produce Pc-NCU06563-gfp.

The analogs *stk-12*(K736Y), *igo-1*(S47A), and *igo-1*(S47D) were generated by site-directed mutagenesis using high-fidelity PCR polymerase, and inserted into pMF272 to produce plasmids Pc-STK-12$^{K736Y}$-gfp, Pc-NcIGO-1$^{S47A}$-gfp, and Pc-NcIGO-1$^{S47D}$-gfp, respectively.

Transformation by electroporation was performed as described previously [63]. Transformants with histidine prototrophy were further confirmed by PCR and GFP fluorescence. To obtain homokaryotic strains, His$^{+}$GFP$^{+}$ transformants were backcrossed to FGSC 9716.

## Microscopy and imaging

Samples were observed under an Olympus BX51 microscope and photographed with a QImaging Retiga 2000R camera (QImaging, Surrey, Canada). Images were analyzed with Image-Pro Express 6.3 software. Samples were also observed under an Olympus SZX-7 stereomicroscope and photographed with the attached digital camera. For nuclear staining, mycelia were stained with 4′,6′-diamidino-2-phenylindole (DAPI) (Sigma-Aldrich) at 1 μg/mL.

For transmission electron microscopy analyses, the mycelia were collected by centrifugation at 2,500 g for 10 min and washed three times with PBS buffer. The harvested mycelia were fixed for 8 h in 0.1 M phosphate buffer containing 2.5% glutaraldehyde at 4˚C, and then for 1 h in 1% osmium tetroxide at room temperature. The samples were gradually dehydrated in an ethanol series and embedded in LR White Resin. Ultrathin sections were stained with uranyl acetate for 30 min and then with lead citrate for 5 min. The stained sections were observed under a Hitachi HT7700 transmission electron microscope (Hitachi, Tokyo, Japan).

## Protein and enzyme activity assays

For protein and enzyme activity assays, 500 μL culture broth was collected at each time point, and then centrifuged at 12,000 g for 10 min to remove mycelia. The supernatant was stored at 4˚C for analysis within 24 h. The protein concentration was determined using a Bio-Rad Protein Assay kit according to the manufacturer's instructions (Bio-Rad, Hercules, CA, USA). As previously described [63], exoglucanase and β-glucosidase activities were assayed at 50˚C

using 1 mg/mL *p*-nitrophenyl-β-D-glucopyranoside (*p*NPG) and *p*-nitrophenyl-β-D-cellobio-side (*p*NPC) as the substrates, respectively. Endoglucanase activity was measured with an azo-CMC kit (Megazyme, Wicklow, Ireland) according to the manufacturer's protocol. Inactive enzyme, which was boiled at 100˚C for 10 min, was used as a control.

## Generation of antiserum against STK-12

*Escherichia coli* BL21 cells and the pGEX-4T-1 plasmid were used for expression of the GST-STK-12 (amino acids T155–C436) fusion protein. The truncated fragment of *stk-12* was cloned by PCR with the primers GST-STK-12-F and GST-STK-12-R, and inserted between the *Bam*H I and *Sma* I sites of pGEX-4T-1 by Gibson Assembly to generate pGEX-STK-12. The resulting plasmid was subsequently introduced into *E. coli* BL21 for protein expression. The purified fusion protein was used as the antigen to generate rabbit polyclonal antiserum as previously reported [64].

## Yeast two-hybrid assay

The yeast two-hybrid system was a gift from Dr. Zhonghai Li, and the plasmids pGBK-ecoI and pGAD-mms22 were gifts from the laboratory of Prof. Huiqiang Lou. The full-length ORF of IGO-1 of *N. crassa* was amplified by PCR using *N. crassa* cDNA as the template. The PCR fragments were cloned into the plasmid pGAD-T7 with the corresponding restriction sites, resulting in AD-IGO-1. The truncated fragment of *stk-12* was cloned into the partner plasmid pGBK-T7 to generate BD-STK-12. The AD-X and BD-STK-12 plasmids were co-transformed into *S. cerevisiae* AH109 and cultured on SD medium without Trp and Leu for 3 days at 28˚C. Protein–protein interaction assays were performed on SD plates without Trp, Leu, and His, and on SD plates without Trp and Leu to avoid artificial results. pGAD-T7/pGBK-ecoI and pGBK-T7/pGAD-mss22 pairs were used as internal negative and positive controls, respectively. To eliminate false positive results, AD-IGO-1 paired with an empty pGBK-T7 was introduced into AH109. Similarly, BD-STK-12 paired with an empty pGAD-T7 was introduced into AH109.

## RNA extraction and RNA sequencing

For RNA-seq experiments, conidia of the *N. crassa* strains were inoculated at $10^5$ conidia/mL into 100 mL 2% w/v Avicel VMM and grown at 25˚C in constant light and shaking at 200 rpm. Mycelia were collected by vacuum filtration at indicated time points (12, 24, 72 and 120 h) and immediately frozen in liquid nitrogen. The frozen mycelia were ground in liquid nitrogen using a mortar and pestle. Total RNA was isolated from frozen samples using Trizol reagent (Invitrogen, Carlsbad, CA, USA), and further treated with DNase I (RNeasy Mini Kit, Qiagen, Hilden, Germany) according to the manufacturer's guidelines. The RNA integrity was checked by agarose gel electrophoresis. RNA-seq was performed on the Illumina HiSeq platform of Novogene (Tianjin, China). All data in this study were generated by sequencing two independent duplicate samples. Prior to read mapping, adaptors and low-quality reads were removed according to Novogene (Tianjin, China) standard protocols. Filtered clean reads were mapped against predicted transcripts from the *N. crassa* OR74R genome v12 with TopHat (version 2.0.12) [65], and the output bam files were stored with SAMtools (version 0.1.19) for subsequent analysis [66]. The number of reads that uniquely mapped to only one gene was calculated by HTseq-count (version 0.6.0) using bam files and genome annotation data as inputs. The abundance of each transcript was calculated from the reads per kilobase per million (RPKM) values [67]. Significant differential expression between WT and Δ*stk-12* was determined using NOISeq (version 2.6.0) [68] (Q value≥0.90 as the threshold, which

approximately corresponds to a |log2 ratio|≥1). To discover significantly up-regulated and down-regulated genes between WT and Δ*stk-12*, only genes with relatively high transcript abundance (RPKM value >20 in at least one strain) were considered for further analysis [69]. Profiling data are listed in S3 Table. The FungiFun2 online resource tool (https://sbi.hki-jena. de/fungifun/fungifun.php) was used in functional enrichment analysis [70]. Principle component analysis was performed using the DESeq package in R (3.4.4) (https://www.r-project.org). The sequence data produced in this study can be accessed at the Gene Expression Omnibus (GSE129410).

## Quantitative real-time PCR

Quantitative real-time PCR was performed using a CFX96 real-time PCR detection system (Bio-Rad) with reagents from Toyobo (One-step qPCR Kit) according to the manufacturer's instructions. In each reaction, 50 ng RNA was used as the template. Unless otherwise noted, all RT-qPCR were performed in triplicate with *Actin* (NCU04173) as the endogenous control. The corresponding primers are listed in S5 Table.

## Western blotting and co-immunoprecipitation assays

Mycelia grown in 2% w/v Avicel VMM at 25˚C for 48 h were harvested, ground to a powder in liquid nitrogen, and suspended in buffer (50 mM HEPES [pH 7.4], 137 mM NaCl, 10% glycerol and 1:100 protease inhibitor [BestBio, Shanghai, China]). The samples were mixed thoroughly by vortexing, and then incubated on ice for 10 min. The samples were centrifuged at 8,000 g for 15 min to remove debris, and the supernatant was collected for subsequent analysis. Protein extracts were loaded on a NuPAGE 4–12% Bis-Tris gel (Life Technologies, Palo Alto, CA, USA). Electrophoresis was performed at a constant voltage of 200 V for 60 min at room temperature. Proteins were transferred to a PVDF membrane in transfer buffer (384 mM glycine, 50 mM Tris-HCl, 20% v/v methanol) at 150 mA for 4 h using a Bio-Rad Mini Trans-Blot Cell. After transfer, the PVDF membrane was rinsed with methanol for 1 min, immersed in PBST (PBS containing 0.3% Tween 20; 20×PBS: 160 g/L NaCl, 28.8 g/L $Na_2HPO_4$, 4 g/L KCl) for 3 min, and then incubated with the primary antibodies in PBST containing 5% (w/v) non-fat dry milk for 1 h. After washing three times with PBST, the membrane was incubated with the HRP-conjugated secondary antibody for 30 min at room temperature. The membrane was washed three times with PBST for 5 min, and then stained using a DAB kit (CWBIO, Beijing, China). The anti-STK-12 polyclonal antibody was diluted in 1:1000. Other antibodies were diluted according to the manufacturer's recommendations.

For Co-IP assays, the *igo* mis-expression strain Pc-NcIGO-1 was inoculated into 2% w/v Avicel VMM and grown at 25˚C in constant light with shaking (200 rpm) for 2 days. The mycelia were harvested by vacuum filtration and immediately frozen in liquid nitrogen. Total proteins were isolated as described above. The samples were incubated with the anti-STK-12 antibody for 4 h at 4˚C, and then incubated with protein A or G magnetic beads (Thermo Scientific) according to the manufacturer's instructions. Proteins eluted from beads were analyzed by western blotting with anti-STK-12 and anti-GFP antibodies (Abmart, Shanghai, China) as described above. To avoid signals from denatured IgG, clean-blot IP detection reagent (Thermo Scientific, Waltham, MA, USA) was used as the secondary antibody.

## Deleting *Mtstk-12* in *M. thermophila*

To construct the plasmid for *Mtstk-12* disruption, the 5' end and 3' end of *Mtstk-12* were cloned by PCR and inserted into the *EcoR*I and *Spe*I sites, respectively, of pPK2BarGFPD [71].

The disruption mutant was obtained utilizing *Agrobacterium tumefaciens* AGL-1 as described [72].

## Statistical analyses

Unless otherwise noted, all experiments were performed in triplicate. Statistical significance was determined by two-tailed Student's *t* test.

## Supporting information

**S1 Fig. Fold change in transcript levels of corresponding genes in wild-type (WT) strain when exposed to different carbon sources in switch experiments.** Gene transcript levels were normalized to 1 when induced with 2% (w/v) glucose. *Actin* (NCU04173) was used as the control. Transcript abundance was evaluated by quantitative real-time PCR. Values represent means of at least three biological replicates. Asterisks indicate significant differences from control (**, $P < 0.01$; ***, $P < 0.001$) based on two-tailed Student's *t*-test.
(JPG)

**S2 Fig. Germination rate of wild-type (WT) and Δ*stk-12* strains on 2% Avicel.** Conidia of Δ*stk-12* and WT were separately inoculated into 100 mL 2% w/v Avicel VMM at $10^6$ conidia/mL and grown at 25°C in constant light with shaking (200 rpm). Germination rate was recorded at 6 h after inoculation. Circles indicate values of individual biological replicates. Error bars show standard deviation. Statistical significance was determined using two-tailed Student's *t*-test (**, $P < 0.01$).
(JPG)

**S3 Fig. Biomass accumulation of WT and Δ*stk-12* mutant when grown on Avicel medium.** Conidia from Δ*stk-12* and wild type (WT) strains were inoculated into Avicel medium, respectively, and batch cultured for 6 days. The biomass accumulation was measured. Values represent the means of at least four replicates, error bars show standard deviation.
(JPG)

**S4 Fig. Principal component analysis (PCA) of RNA-seq data for wild-type (WT) and Δ*stk-12* strains grown on Avicel.**
(JPG)

**S5 Fig. Deletion of *stk-12* results in increased secretion of hydrolytic enzymes. Total extracellular protein concentration (A) and endoglucanase activity (B).** Values represent the means of at least three replicates, error bars show standard deviation.
(JPG)

**S6 Fig. Stability of mRNA is not altered by deletion *stk-12*.** The decay of the *cbh-1* mRNAs in the WT and *stk-12* mutant is shown at the indicated time points after addition of thiolutin. Conidia from Δ*stk-12* and wild type (WT) strains were inoculated into Avicel medium, respectively, and batch cultured for 1 day. And then, thiolutin was added to a final concentration of 12 μg/mL to stop transcription. CBH-1 mRNA levels were measured by RT-qPCR and the levels of mature 26S rRNA was used as the internal control.
(JPG)

**S7 Fig. Quantitative RT-PCR of target genes expression levels in WT and overexpression strains.** All strains were grown in MM for 16 h, and then transferred into Avicel medium for an additional 4 h. Gene expression levels were measured by RT-qPCR using actin as a control

and normalized against the tested gene, *igo-1* (A) or *stk-12* (B), in WT strain.
(JPG)

**S8 Fig. Subcellular localization of STK-12-GFP.** The strain with *stk-12* under control of the native promoter (Pn-STK-12) was grown on MM plates for 5 days. Scale bar = 20 μm.
(JPG)

**S9 Fig. Phylogenetic analysis of STK-12 and its homologs conducted using MEGA 6 software.** Bootstrap values are adjacent to each internal node (% of 1,000 bootstrap replicates). NCU04566 (PRK-10; SNF 1 homolog) from *Neurospora crassa* was used as outgroup.
(JPG)

**S10 Fig. Phenotype of WT and Δ*igo-2* strains when grown on Avicel medium.** Avicel broth cultures were inoculated with conidia and grown for 5 days. Total extracellular protein concentration and endoglucanase activity of culture broth were measured and are expressed as a percentage of those in WT. Values represent means of at least three biological replicates, error bars show standard deviation. Statistical significance was determined by two-tailed Student's *t*-test (n.s., not significant).
(JPG)

**S11 Fig. Transcript levels of genes encoding major cellulases and sugar transporters in Δ*igo-1* mutant relative to wild-type (WT) strain on Avicel.** After Δ*igo-1* and WT conidia were grown on Avicel for 3 days, transcript abundance of indicated genes was evaluated by quantitative real-time PCR. Values represent means of at least three biological replicates, error bars show standard deviation. Statistical significance was determined by two-tailed Student's *t*-test (\*, $P<0.05$; \*\*, $P < 0.01$; \*\*\*, $P < 0.001$, n. s., not significant).
(JPG)

**S12 Fig. Subcellular localization of PP2A in *Neurospora crassa*.** Strain with NCU06563 under control of *ccg-1* promoter was grown on MM plates for 5 days. Location of PP2A was monitored by GFP fluorescence. Scale bar = 20 μm.
(JPG)

**S13 Fig. Transmission electron micrographs of *N. crassa* wild-type (WT) strain (right) and Δstk-12 mutant (left) after 3 days of culture in minimal medium with 2% (w/v) Avicel as sole carbon source.** Mycelia were collected and prepared for transmission electron microscopy. White arrows indicate endoplasmic reticulum. M, mitochondrion; V, vacuole.
(TIF)

**S1 Table. List of serine/threonine kinase mutants in *Neurospora crassa*.**
(XLSX)

**S2 Table. The expression of 301 Carbohydrate Activity Enzyme (CAZy) genes during submerged cultivation.**
(XLSX)

**S3 Table. Gene expression profiling and differential gene expression analysis of Δ*stk-12* vs. wild-type (WT) grown on Avicel medium for indicated times.**
(XLSX)

**S4 Table. The expression of cellulase genes during submerged cultivation.**
(XLSX)

**S5 Table. Primers used in this study.**
(XLSX)

## Acknowledgments

We thank Huanhuan Zhai for assistance with transmission electron microscopy. We are grateful to Zhonghai Li (Shangdong University) for the gift of the yeast strains, to Huiqiang Lou (China Agricultural University) for the plasmids pGBK-ecoI and pGAD-mms22.

## Author Contributions

**Funding acquisition:** Liangcai Lin, J. Philipp Benz, Chaoguang Tian.

**Investigation:** Liangcai Lin, Shanshan Wang, Xiaolin Li, J. Philipp Benz, Chaoguang Tian.

**Methodology:** Liangcai Lin, Qun He, Chaoguang Tian.

**Supervision:** Chaoguang Tian.

**Writing – original draft:** Liangcai Lin, J. Philipp Benz, Chaoguang Tian.

**Writing – review & editing:** Liangcai Lin, Qun He, J. Philipp Benz, Chaoguang Tian.

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
