## [Decision Letter · Decision Letter 0]

22 Aug 2019

Dear Tian,

Thank you very much for submitting your Research Article entitled 'STK-12 acts as a transcriptional brake to control the expression of cellulase-encoding genes in Neurospora crassa' to PLOS Genetics. Your manuscript was fully evaluated at the editorial level and by independent peer reviewers. The reviewers appreciated the attention to an important problem, but raised some substantial concerns about the current manuscript. Based on the reviews, we will not be able to accept this version of the manuscript, but we would be willing to review again a much-revised version. We cannot, of course, promise publication at that time.

If you decide to revise the manuscript for further consideration at PLOS Genetics, please aim to resubmit within the next 60 days, unless it will take extra time to address the concerns of the reviewers, in which case we would appreciate an expected resubmission date by email to plosgenetics@plos.org.

[LINK]

We are sorry that we cannot be more positive about your manuscript at this stage. Please do not hesitate to contact us if you have any concerns or questions.

Yours sincerely,

Katherine A. Borkovich, Ph.D.

Guest Editor

PLOS Genetics

Gregory P. Copenhaver

Editor-in-Chief

PLOS Genetics

Dear Tian:

Your manuscript has been reviewed by three experts in the field and I have also read the paper. All three reviewers thought the results were of interest and present new information about the regulatory mechanisms that control cellulase enzyme production and secretion in fungi. However, they believe that the manuscript will be improved by some revision and inclusion of additional information. The overall decision is Major Revision.

The major concern, raised by Reviewers 1 and 3, and shared by me, is that no data is presented for the amount of fungal biomass accumulated by wild type and the stk-12 mutant during the time cellulase activity and mRNA levels are being monitored. The rationale is that the larger the biomass, the greater the expression and secretion capacity for these enzymes in a particular strain. This concern is heightened by the observation that the stk-12 mutant germinates faster than wild type in cellulose medium (as shown in the supplementary material) and thus may put on more mass more quickly. The questions about biomass must be addressed in a quantitative manner in a revised manuscript.

Reviewer 2 notes that more time points are needed for the early figures that present the persistence phenotype for stk-12 mutants. Reviewer 2 also expresses concerns about using the heterologous ccg-1 promoter instead of the native promoter to drive expression of the STK-12-GFP protein. Reviewer 3 would like to see data demonstrating that the mutant alleles of stk-12 and igo-1 are actually expressed. Reviewer 3 also notes that new results are presented in the Discussion—these should be moved to the Results section or eliminated from the paper. In addition, the Discussion is too long and repeats much of the results.

Please address the comments of all three reviewers in your revised manuscript.

Reviewer's Responses to Questions

**Comments to the Authors:**

Reviewer #1: The manuscript STK-12 acts as a transcriptional brake to control the expression of cellulase-encoding genes in Neurospora crassa is important to the field and should be considered to be published in Plos genetics . However, may concern is about the phenotypes of mutants. No data about growth is presented. For cellulase the growth of fungi is directly associated to cellulase production. For isntance, if enzyme acitvy data be presented as specific activity, I am not convinced about the regulation of STK-12 claimed by authors.

Another important point: RNAseq experiment is poor explained in methods. It is important to perform the experiments using mycelia transference, since its well know that kinases and phosphatases influence in fungi growing and developing. It will be a pleasure to evaluate the manuscript again if this points raised been addressed.

Reviewer #2: The manuscript by Lin et al described the screening of a protein kinase deletion set in Neurospora crassa and identified a kinase called STK12 which they presented evidence assigning it as a transcriptional brake on cellulase gene expression during a later stage of the inducing process. A key important effector downstream SKT12 in modulating celluase gene expression was thought to be a broad phopsphotase PP2A, the negative regulator of which, IGO1, was found to be a direct substrate of SKT12. Therefore, the aberrant activation of PP2A due to the lack of inhibition from IGO was thought to result in the no back-to-basal level expression of cellulase genes. The results presented here, although provide new insights into the extricate regulatory mechanism of cellulase gene expression in relevant filamentous fungi, still leave quite a few outstnding concerns unaddressed.

1)Considering that cellulase gene mRNA levels in stk12 were maintained for a significant longer time than other mutants, possibility exists that these mRNAs might be just more stable due to the absence of skt12. The authors should check this.

2)A relevant question to the above point is that, since skt12 is not required for the initial induced cellulase gene expression but involved in a later backup inhibitory process, precisely determining the kinetics of SKT1 activation as well we its substrate phosphorylation and so on will be key to elucidating the involved mechanisms.

3)Fig. 1B-I, transcriptional kinetics based on more time points should be determined. In this way, for example, the exact persisting time for the induction in the skt1 mutant could be more precisely shown and compared with others.

4)Since the data finally point to PP2A as being required for cellulase induction, the authors should present evidence whether IGO phosphorylation by SKT12 indeed failed to inhibit PP2A, and timely phosphorylation and thus activation of IGO as well as the corresponding inhibition of PP2A is indeed necessary for the bell-shaped cellulase gene expression.

5)Is PP2A activity also required for the initial induction? Does there exist an opposite kinetic pattern of PP2A activation relative to that of STK12?

6)For cellular localization, overexpression may cause a mislocalization and SKT should be driven by its own promoter instead of ccg-1 promoter. And how about the cellular localization of IGO and PP2A?.

7）The kinase mutant K736Y should be verified to be expressed appropriately, more importantly the kinase activity should be tested in vitro.

8）The transcriptional kinetics of SKT12 itself should be determined considering that overexpression of skt1 interferes with cellulase expression as well as development.

9) The fold change in deltaskt1igo1 was more dramatic than that of individual single deletion mutant, which means there was probably a synergistic effect between the deletion of skt1 and igo1. The authors might want to introduce the IGO S47D into the single delta igo1 and skt1 deletion strain, respectively, to see its inhibitory effect.

10) Since IGO and PP2A were thought to be the important downstream components of the STK12 pathway, it is kind of weird that no general information about function og these two proteins was presented in the introduction.

11) “Overall, the transcriptome data and transcriptomic analyses suggested that deletion of stk-12 increased the time required to activate RESS. Increased cellulase production is therefore most likely due to long-term maintenance of high cellulase gene expression levels, but not to an increase in their transcript levels” lacks solid evidence.

12)“These results indicate that phosphorylation by STK-12 is essential for IGO-1 activation, and also imply that the corresponding phosphorylation event is conserved between yeast and filamentous fungi. ”is premature conclusion wihthout solid evidence.

13）The part about the changes in ER structure does not seem to fit anywhere in the whole story.

Other comments:

1)Line numbers should be included for review.

2)Figure labels are some kind of blurred. Need to be adjusted.

3)“The extracellular protein concentration of the Δigo-1 strain was increased by approximately 4.3-fold, and endoglucanase activity was increased by about 3.8-fold compared with the WT”. This is not true, the protein content increased only 2-fold while there was no endoglucanase activity in Fig. 6A.

Reviewer #3: In their manuscript the authors describe investigation of a kinase influencing cellulase regulation as well as parts of the regulatory mechanism in this pathway. The study is interesting in general and important in the field. Experimental design is largely sound and the manuscript is well written, although some modifications could benefit the manuscript. The main problem of the study is, that no data on specific cellulase activities are provided, just activities per ml in supernatants. However, it is of utmost importance to consider biomass production of the fungus in such a study. The differences in activity can very well be due to biomass alterations alone. On the other hand, if the deletion of the kinases investigated causes a growth defect on cellulose, the beneficial effects due to lack of the respective gene could even be much larger than presented here. Therefore, it is mandatory for studies on cellulase production to provide results related to biomass and the data are incomplete in the current form.

Besides this problem, I have some comments to improve the manuscript as listed below.

The manuscript lacks line numbers which makes reviewing very inconvenient. This must be corrected in the revised version.

On page 7 – still in the introduction section, the authors describe already some results. If these results were obtained in a previous study, then this study must be cited. If not, please reorganize to describe results in the Results section.

Page 10

The higher cellulase production on day 3 levels out at later time points – no not only a prolongued induction period should be considered, but an earlier onset as well.

Specify what you precisely mean by lignocellulolytic enzyme production – activities (which ones?), abundance of enzymes (proteins analyzed by Western blots) or else.

Page 12

Enriched means that more genes of a certain function are in the respective gene set than they are in the entire genome. An individual gene cannot be enriched, only functions can be enriched in a gene set. Reword accordingly.

Xyr-1 is also important in cellulase regulation in N. crassa – please mention its regulation here as well.

Do these data correlate with phosphorylation patterns found by Xiong et al., 2014? It should be considered, that altered regulation of such a transcription factor gene, indicates that the phosphorylation state of an upstream (transcription) factor was altered, which caused this change.

Page 12, first line

Twofold is not a dramatic change, maybe 10fold but not less.

For the other regulations mentioned on this page, specify fold regulations and standard deviations or p-values compared to wildtype.

For the uptake system, provide more precise numbers – for example in case of an only twofold downregulation, an influence could be mentioned, but the system would not be “compromised”.

Page 14

...cellulase gene expression levels and not only to an increase in their transcript levels.

Both regulation mechanisms are important!

Page 17 and elsewhere

RT-qPCR

Actin was used as the only reference gene. Please confirm that this reference gene was checked for its stability under the conditions applied in the study (using the software geNorm or the like). According to MIME criteria, one reference gene is not sufficient and several genes at least have to be tested to select for stability.

The discussion section contains description of additional results. These should be moved to the resultls section and be described there. Then the authors can refer to these results in the discussion.

Page 28

Was the misexpression confirmed by RTqPCR? Only if the transcript abundance is indeed altered, then the results can be considered relevant.

Please also specify how many misexpression mutants were used and if the results were consistent – also considering misexpression levels.

Page 31

Enzyme stability is an important issue in cellulase analysis, especially at 4°C. Please specify how long after harvesting the enzyme analysis was done and if controls were done at the same time.

Figure 1B-I

Specify which gene’s expression (or rather transcript levels!) is shown here.

Figure 12

This figure should be more precise, showing an influence on gene regulation (transcript abundance), which is not confused with protein abundance, which was not analyzed. Does STK-12 have an influence on clr-1? The text in the results says otherwise and the impact on CLR-2 cannot be explained this way. In its current version the model contains mistakes and lacks precision. Modify.

**Have all data underlying the figures and results presented in the manuscript been provided?**

Reviewer #1: Yes

Reviewer #2: Yes

Reviewer #3: Yes

PLOS authors have the option to publish the peer review history of their article (what does this mean?). If published, this will include your full peer review and any attached files.

Reviewer #1: No

Reviewer #2: No

Reviewer #3: No

---

## [Decision Letter · Decision Letter 1]

5 Nov 2019

Dear Dr Tian,

We are pleased to inform you that your manuscript entitled "STK-12 acts as a transcriptional brake to control the expression of cellulase-encoding genes in Neurospora crassa" has been editorially accepted for publication in PLOS Genetics. Congratulations!

Please note Reviewer #3 has a minor suggestion (see below) that you can address as you prepare your final draft for the production team (the editorial team will not need to reevaluate).

Yours sincerely,

Katherine A. Borkovich, Ph.D.

Guest Editor

PLOS Genetics

Gregory P. Copenhaver

Editor-in-Chief

PLOS Genetics

Comments from the reviewers (if applicable):

Reviewer's Responses to Questions

Comments to the Authors:

Please note here if the review is uploaded as an attachment.

Reviewer #1: Once the biomass of Δstk-12 and WT were determined during the cultivation and no difference was observed, all data are validated to take authors conclusions and inferences.

Reviewer #3: The authors have carefully revised the manuscript according to my comments and those of the other reviewers. I consider it now appropriate for publication.

However, there seems to be a misunderstanding in the abbreviation of quantitative reverse transcriptase polymerase chain reaction. The common abbreviation is RT-qPCR, while RT-PCR is mostly used for the method involving only gel electrophoresis for visual estimation of signal strength. This should be corrected to avoid confusion.

Have all data underlying the figures and results presented in the manuscript been provided?

Large-scale datasets should be made available via a public repository as described in the 

PLOS Genetics

data availability policy, and numerical data that underlies graphs or summary statistics should be provided in spreadsheet form as supporting information.

Reviewer #1: Yes

Reviewer #3: Yes

PLOS authors have the option to publish the peer review history of their article (what does this mean?). If published, this will include your full peer review and any attached files.

Do you want your identity to be public for this peer review?

 For information about this choice, including consent withdrawal, please see our Privacy Policy.

Reviewer #1: No

Reviewer #3: No

**Data Deposition**

http://datadryad.org/submit?journalID=pgenetics&manu=PGENETICS-D-19-01211R1

Press Queries

---

## [Editor Report · Acceptance letter]

19 Nov 2019

PGENETICS-D-19-01211R1 

STK-12 acts as a transcriptional brake to control the expression of cellulase-encoding genes in Neurospora crassa 

Dear Dr Tian, 

We are pleased to inform you that your manuscript entitled "STK-12 acts as a transcriptional brake to control the expression of cellulase-encoding genes in Neurospora crassa" has been formally accepted for publication in PLOS Genetics! Your manuscript is now with our production department and you will be notified of the publication date in due course.

With kind regards,

Kaitlin Butler

PLOS Genetics

On behalf of:
